# A Copernicus-based evapotranspiration dataset at 100-m spatial resolution over four Mediterranean basins

Paulina Bartkowiak[1], Bartolomeo Ventura[1], Alexander Jacob[1], Mariapina Castelli[1]

[1]Insitute for Earth Observation, EURAC Research, Bozen-Bolzano, 39110, Italy

*Correspondence to*: Paulina Bartkowiak (paulina.bartkowiak@eurac.edu)

**Abstract.** Evapotranspiration (ET) is responsible for regulating the hydrological cycle with a relevant impact on air humidity and precipitation, particularly important in the context of acute drought events in recent years. With the intensification of rainfall deficits and extreme heat events, the Mediterranean region requires regular monitoring to enhance water resources management. Even though remote sensing provides spatially continuous information for estimating ET on large scales, existing

global products with spatial resolution ≥ 0.5 km are insufficient to capture spatial detail at a local level. In the framework of the ESA 4DMed-Hydrology project, we generate an ET dataset at both high spatial and temporal resolutions by the Priestley-Taylor Two-Source Energy Balance model (TSEB-PT) driven by Copernicus satellite data. We build an automatic workflow to generate 100-m ET product by combining data from Sentinel-2 (S2) MSI and Sentinel-3 (S3) land surface temperature (LST) with ERA5 climate reanalysis derived within the period 2017-2021 over four Mediterranean basins in Italy, Spain,

France, and Tunisia (Po, Ebro, Hérault, Medjerda). First, original S2 data are pre-processed before deriving 100-m inputs for the ET estimation. Next, biophysical variables, like leaf area index, and fractional vegetation cover are generated, and then they are temporally composited within a 10-day window according to Sentinel-3 acquisitions. Consequently, decadal S2 mosaics are used to derive the remaining TSEB-PT inputs. In parallel, we sharpen 1-km S3 by exploiting dependency between coarse-resolution LST and 100-m S2 reflectances using the decision trees algorithm. Afterward, climate forcings are utilized

for modeling energy fluxes, and next for daily ET retrieval. The daily ET composites demonstrate reasonable TSEB-PT estimates. Based on the validation results against eight Eddy Covariance (EC) towers between 2017-2021, the model predicts 100-m ET with an average root mean square error of 1.38 mm day$^{-1}$ and Pearson coefficient equal to 0.60. Regardless of some constraints, mostly related to the high complexity of EC sites, TSEB-PT can effectively estimate 100-m ET, which opens up new opportunities for monitoring the hydrological cycle on a regional scale. The full dataset is freely available at

https://doi.org/10.48784/b90a02d6-5d13-4acd-b11c-99a0d381ca9a,        https://doi.org/10.48784/fb631817-189f-4b57-af6a-38cef217bad3,  https://doi.org/10.48784/70cd192c-0d46-4811-ad1d-51a09734a2e9,  and  https://doi.org/10.48784/7abdbd94-ddfe-48df-ab09-341ad2f52e47 for Ebro, Hérault, Medjerda, and Po catchments, respectively (Bartkowiak et al., 2023a-d).

# 1 Introduction

Terrestrial evapotranspiration is a keystone component for estimating water loss from the Earth's surface, being the main indicator of biophysical conditions for vegetation and bare soil (Coenders-Gerrits et al., 2020; Gouveia et al., 2017). Designated as an essential climate variable, ET substantially contributes to hydrological, energy, and carbon cycles through its high sensitivity to atmosphere-land interactions, particularly crucial in the context of a warming climate (Fisher et al., 2017; Konapala et al., 2020). Given its significance, ET finds widespread use in various environmental applications, including
climate studies (Chattopadhyay and Hulme, 1997; Dezsi et al., 2018; Gao et al., 2017), drought detection (Maes and Steppe, 2012; Otkin et al., 2016), sustainable agriculture and food production (Allen et al., 1998; Cammalleri et al., 2014; Dari et al., 2022), and also natural ecosystem monitoring (Anderson et al., 2012; Granata et al., 2020). However, most of these activities require spatially continuous ET data with regular revisit time and long-term observation records (Jiang et al., 2021). While conventional *in-situ* measurements such as lysimeters, Eddy Covariance, and Bowen ratio techniques (Allen et al., 1991; Buttar
et al., 2008; Pastorello et al., 2020) provide frequent time-series, their limited spatial coverage and sparse network restrict their practical utility compared to gridded ET products.

Over the past few decades, many methods have been developed for retrieving spatially distributed ET over large areas, which can be categorized into two groups: process-based and data-driven approaches. The first category comprises physical modeling methods that derive energy fluxes based on theoretical assumptions. These approaches rely on various forcing
parameters to explain ET and have been proposed by many researchers (Allen et al., 1998; Monteith, 1965; Norman et al., 1995; Penman, 1948; Priestley and Taylor, 1972; Su, 2002; Mallick et al., 2014). In addition to the "single-pixel methods" (Chirouze et al., 2014), LST-based contextual methods of ET that calibrate energy balance at dry/hot and wet/cold conditions within an image have been successfully applied in numerous studies (Bastiaanssen et al., 2005; Sobrino et al., 2021; Trezza et al., 2013). In contrast, the second group is based on empirical relationships between ET and its controlling predictors. These
relationships are derived from *in-situ* and remotely sensed observations and are mainly established using statistical regressions (Maselli et al., 2014), machine learning (ML) and deep learning (DL) algorithms, like random forest (Douna et al., 2021), artificial neural networks (Ferreira et al., 2019; Jain et al., 2008), and long short-term memory (Babaeian et al., 2022). In these data-driven approaches, the focus is primarily on the statistical patterns and correlations between the observed variables and ET, with minimal incorporation of physical mechanisms into a model. In recent years, to achieve a balance between physical
principles and model-learned relationships over large and diverse datasets, the scientific community has proposed process-constrained ML/DL methods that combine data-driven algorithms with process-based modeling (Cui et al., 2021; Hu et al., 2021; Reichstein et al., 2019).

Regardless of the ET method employed, open-source global climate datasets, like the European ReAnalysis V5 (ERA5) and NASA Global Land Data Assimilation System (GLDAS), in conjunction with a surge in spaceborne Earth Observation
(EO) technologies have greatly accelerated development of many gridded ET products (Bhattarai and Wagle, 2021; García-Santos et al., 2022). Although ET cannot be directly measured from space, since the 1980s EO satellites have been providing

valuable observations of the land surface parameters, enabling estimation of actual ET at large scales (Zhang et al., 2016). One notable advantage of freely accessible global ET products is their spatiotemporal continuity and related long-term availability. In recent years, several long-term datasets have been developed, including the GLDAS Catchment Land Surface Model
(CLSM) encompassing the years 1948 to 2014, and the Priestley–Taylor-based Global Land Evaporation Amsterdam Model (GLEAM), covering the period from 1980 to the present (Li et al., 2018; Miralles et al., 2011; Martens et al., 2017). Although these global-scale models provide extensive time-series at the continental level, serving as a valuable parameter for many hydrological models (Alfieri et al., 2022; López López et al., 2017) and offering important insights into water availability at a large scale (Bai and Liu, 2018), they provide ET data at spatial resolutions of tens of kilometres. Since 2015, daily 3-km ET
maps, driven by the Spinning Enhanced Visible and InfraRed Imager (SEVIRI) onboard the geostationary Meteosat Second Generation (MSG) satellite, have been generated and made available through the LSA-SAF system (*https://landsaf.ipma.pt/*). Additionally, other global ET products include the Penman–Monteith MOD16 maps driven by MODerate resolution Imaging Spectroradiometer (MODIS) inputs (Mu et al., 2007), and Operational Simplified Surface Energy Balance (SSEBop) ET, typically utilizing the 1-km MODIS LST and Leaf Area Index (LAI) products and climate reanalysis datasets (Senay et al.,
2013), among others that have been successfully applied in many regions (Weerasinghe et al., 2020). Despite their reliable validation results in relatively homogenous landscapes, like the contiguous United States (CONUS) with an average $R^2$ of 0.7 for both SSEBop and MOD16 (Velpuri et al., 2013), their large pixel size and related insensitivity to complex terrain might not be representative over heterogenous locations (Castelli, 2021; McShane et al., 2017).

Surface energy balance (SEB) modeling is a valuable tool for estimating ET using high spatial resolution (HR) thermal
remote sensing, like sub-field scale Landsat LST imagery with a pixel size ranging from 60-120 m, and 70-m ECOsystem Spaceborne Thermal Radiometer Experiment (ECOSTRESS) mission launched in 2018 (Anderson et al., 2021; Xue et al., 2022). Currently, the global HR ECOSTRESS ET data is generated from the Priestley-Taylor Jet Propulsion Laboratory (PT-JPL) algorithm (Fisher et al., 2008; Fisher, 2018). To expand the high capabilities of the HR LST, Cawse-Nicholson and Anderson (2021) have introduced the disaggregated Atmosphere-Land Exchange Inverse Jet Propulsion Laboratory
(DisALEXI-JPL) model, which provides ECOSTRESS-driven energy fluxes over the CONUS area. Additionally, the ESA European ECOSTRESS Hub data repository has been released that offers an open-source 70-m daily evaporation product for Europe and Africa based on the non-parametric Surface Temperature Initiated Closure (STIC) model (Hu et al., 2022; Mallick et al., 2014). Even though the HR ET datasets obtain satisfactory results, the irregular revisit time of ECOSTRESS over Europe and the 8–16-day repeat cycle for Landsat hamper their use in monitoring ET dynamics and temporal trends. In this regard,
next-generations of HR thermal missions are designed, including ESA Copernicus Land Surface Temperature Monitoring (LSTM), Thermal infraRed Imaging Satellite for High-resolution Natural resource Assessment (TRISHNA) of CNES-ISRO (France-India), and NASA Surface Biology and Geology (SGB) TIR. Even though these instruments are planned to be launched between 2024 and 2028, their operational use will be further delayed. Thus, there is an urgent need to bridge this gap by advancing current satellite-based ET estimates.

Given the limited availability of high spatiotemporal LST data, such as those with both daily revisit time and sub-kilometre pixel size, SEB-based retrievals have been commonly enhanced by spatially downscaling daily TIR images obtained from 1-km satellite sensors like Terra/Aqua MODIS, Visible Infrared Imaging Radiometer Suite (VIIRS) onboard Suomi National Polar-orbiting Partnership (S-NPP), and Sentinel-3 Sea and Land Surface Temperature Radiometer (SLSTR) (Bisquert et al., 2016; Guzinski and Nieto, 2019; Xue et al., 2021). Due to the increasing volume and variability of geospatial data, many data-

driven approaches have been proposed, relying on empirical relationships between 1-km surface temperatures and high-resolution explanatory variables derived from Synthetic Aperture Radar (SAR) and Visible Shortwave Infrared (VSWIR) sensors (Amazirh et al., 2019; Li et al., 2019; Mao et al., 2021; Pu and Bonafoni, 2023). For instance, Liu et al. (2020) employed a random forest algorithm to derive 250-m MODIS LSTs over northern China, reporting an improvement in RMSE of 32%-36% compared to the original 1-km images. Furthermore, since 2019 the FAO initiative WaPOR has provided 10-day ET

composites over Africa and the Middle East derived from the ETLook model and driven by MODIS, PROBA-V, and Landsat data at continental, country, and subnational scales, corresponding to resolutions of 250-m, 100-m, and 30-m, respectively (Bastiaanssen et al., 2012; Blatchford et al., 2020). In parallel, the Priestely-Taylor Two-Source Energy Balance model, forced by ESA Copernicus data, has demonstrated potential for producing HR ET with global coverage (Bellvert et al., 2020; Guzinski et al., 2020; Chintala et al., 2022). The frequent acquisitions of the HR Sentinel-2 MultiSpectral Instrument (MSI) and 1-km

Sentinel-3 SLSTR with the two-satellite configurations, along with global ERA5 climate data, serve as reliable inputs for TSEB-PT due to their long-term continuity and evolution plans. As reported by *Guzinski et al.* (2021), Copernicus datasets, including downscaled Sentinel-3 LST to resolutions ranging from 20 to 300 m, exhibit better spatial-scale consistency than WaPOR inputs, resulting in a correlation coefficient equal to 0.9 and a mean bias of less than 0.3 mm day$^{-1}$over Mediterranean agricultural areas in Tunisia and Spain during growing season in 2018-2019.

In this study, we aim to produce Copernicus-based ET maps for the Mediterranean region (MR) utilizing the Two-Source Energy Balance model with a dual-source scheme, which allows estimating energy fluxes for both vegetation and soil components. The ET product covers the years 2017-2021 and is generated at high spatiotemporal resolutions of 100 m on a daily basis as a reasonable scale over a fragmented Mediterranean landscape. The maps are derived using freely available algorithms developed within the ESA Sentinels for Evapotranspiration (Sen-ET) initiative (DHI-GRAS, 2020). In general, we

synergistically combine high-resolution Sentinel-2 shortwave data, moderate-resolution Sentinel-3 LST images, and ERA5 climate observations to generate the ET grids. Due to big data volume and the multi-step processing involved, the objective of this study is also to automatize the entire workflow for large-scale applications and provide recommendations for facilitating Sen-ET inputs and algorithms through cloud computing infrastructure. To achieve this, we implement the entire workflow using cloud computing units offered by VMware, EODC, and CloudFerro. The processing pipelines are designed to update

the resulting ET time-series and make it more suitable for operational use. To the best of our knowledge, it is the first application of the TSEB-PT at sub-kilometre spatial resolution over the Mediterranean Basin. Notably, such areas are often underrepresented in global-oriented studies, making this work particularly useful in advancing our understanding of ET in regions of both high ecohydrological and socio-economic importance.

## 2 Study area and datasets

### 2.1 Study sites

Our study focuses on four Mediterranean river basins: Ebro in Spain, Po in Italy, Medjerda in Tunisia, and the French Languedoc-Roussillon with the Hérault basin. The regions cover a total area of approximately 190 000 km². Figure 1 depicts the geographical locations of all areas of interest.

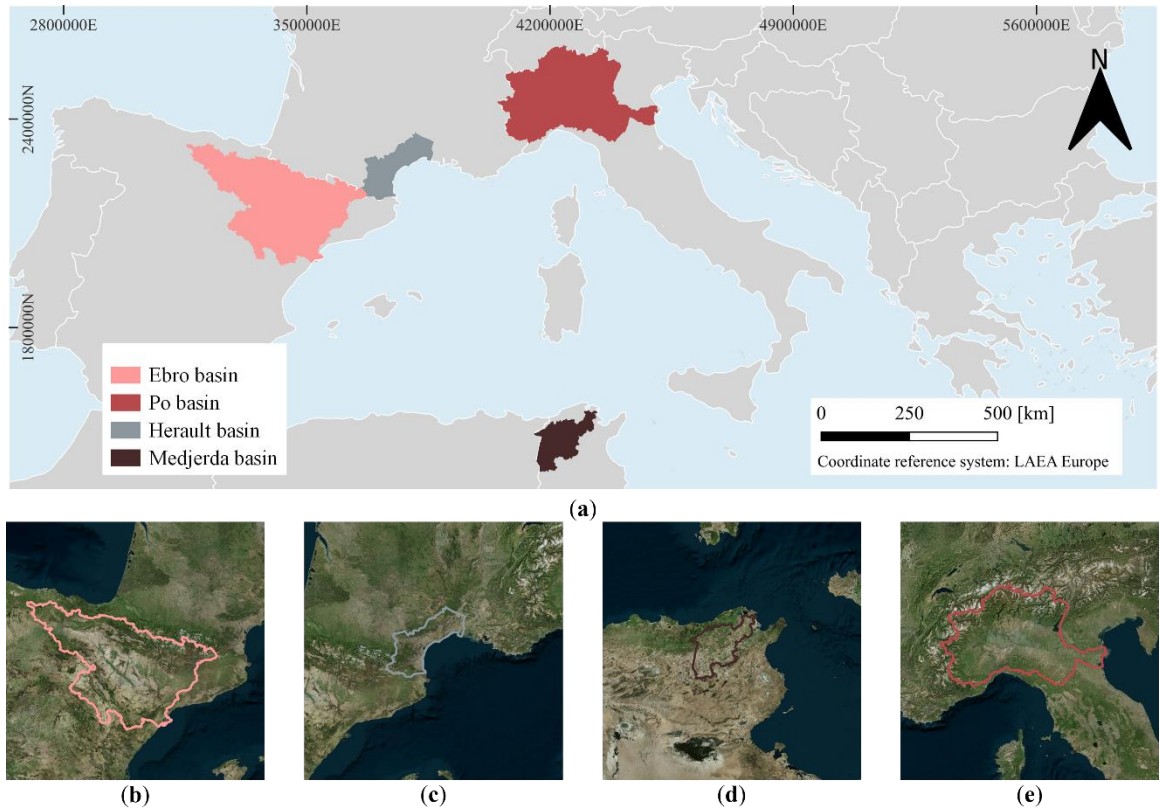

**Figure 1: Overview of the areas of interest: (a) general location of the basins in the Mediterranean region; (b)-(e) Ebro, Hérault, Medjerda, Po basins. All figures were generated in QGIS using internal resources (Figure 1a) and open-source layers provided by *https://www.bing.com/maps* (Figures 1b-e) © Microsoft.**

As demonstrated in Fig.1, the Ebro basin (EB), with a total catchment size of about 85 500 km², exhibits a diverse landscape, leading to varied climatic conditions. The EB experiences an oceanic and Mediterranean mountain climate in the north with an average annual air temperature ($TA_{mean}$) ranging from 9°C to 12°C. Moving southeast, the climate transitions into a warm oceanic climate, where $TA_{mean}$ ranges from 11 to 14°C (Lorenzo-González et al., 2023). Precipitation patterns also vary across the region. The south-eastern part experiences low precipitation (350–700 mm yr$^{-1}$), while the mountain regions receive more rainfall yielding 800-2000 mm per year (Gaona et al., 2022). To meet the water demands of the entire EB, extensive canal systems have been developed, and around 92% of the total water consumption is allocated to irrigation and farming (Barella-Ortiz et al., 2023). Similarly, in the Po basin (PB), which spans an approximate surface area of 71 000 km²,

the climate and water resources are strongly influenced by the region topography. The northern PB is known for its Alpine climate, and it is characterized by numerous water reservoirs commonly used for energy generation. The southern areas, despite having a lower water supply, play a vital role in agriculture due to their large water storage capacities and favourable terrain (Dari et al., 2023). $TA_{mean}$ in the Po basin varies across the region. In the mountains, $TA_{mean}$ ranges from 5 to 10°C, while in the remaining zones it falls within the range of 10 to 15°C (Musolino et al., 2017). Additionally, the average precipitation ranges between 700 and 1500 mm per year (Filippucci et al., 2022). The coastal Languedoc-Roussillon (LR) region with Hérault basin (HB) in southern France represents another agriculture-oriented basin, with nearly 30% of arable land represented by vineyards (Cambrea et al., 2020). The LR is primarily influenced by a Mediterranean climate characterized by hot, dry summers and mild winters. The southern sections of the Pyrenees exhibit nival climatic conditions. Similar to the EB and PB, the HB follows a climatic gradient that is dependent on its geolocation. In the northern part, the yearly $TAs_{mean}$ ranges around 8°C, while precipitation reaches 1600 mm per year. In contrast, the southern part experiences higher temperatures, with $TA_{mean}$ surpassing 15°C, and lower rainfall levels of around 600 mm yr$^{-1}$ (Fabre et al., 2015). The Medjerda basin (MB) in Tunisia represents a catchment with drier and hotter climatic conditions compared to the other study sites. In fact, it is characterized by low annual precipitation varying between 350 mm and 600 mm, coupled with high annual temperatures averaging between 16 and 22°C (Rajosoa et al., 2022). With an area of approximately 15 500 km$^2$, the MB is the largest watershed in Tunisia, and thus it holds significant importance in terms of water supply for both domestic use and farming (Boulmaiz et al., 2022). Indeed, agriculture plays a dominant role, consuming the largest amount of water resources, accounting for nearly 76% of the total water volume available (FAO, 2020).

Given the hydro-demanding activities across the study sites, which include crop irrigation, energy production, mass tourism, as well as domestic water use, and on the other side the challenges posed by extreme heatwaves and recurring droughts in recent years, daily ET maps at the river basin scale would be highly beneficial for supporting vegetation monitoring and sustainable water management practices (Gouveia et al., 2017).

## 2.2 In-situ measurements

In this study, Eddy Covariance measurements collected in the framework of the European Fluxes Database Cluster (EFDC) are used to validate gridded ET products (Heiskanen et al., 2022). EFDC is the European initiative that gathers and standardizes *in-situ* fluxes from a wide range of the EC networks (e.g., ICOS, InGOS, CarboItaly, and GHG-Europe) to facilitate their application among the scientific community worldwide. The database stores long-term measurements (since 1996) that are pre-processed and quality-controlled by providers before data submission to the system.

In this work, after deriving all measurements from the EFDC, all available EC records have been harmonized and made available as a multi-year file stack with daily time step through the project-dedicated PostgreSQL database (*https://edp-portal.eurac.edu/*). Consequently, ground-based EC data are analysed for eight stations located in the study subdomains (see Sect. 2.1 for more details). The sites are represented by different vegetated land covers which include alpine grasslands, forest, and vineyard ecosystems. As depicted in Table 1, five towers are located in mountain regions above 1400 m a.s.l. and are

covered by grasslands and forest, while three remaining towers lie in forest and vineyard biomes in France and Italy at altitudes

ranging from 1 to 270 m a.s.l. Considering the time span of the generated ET product, *in-situ* measurements temporally overlap

with Copernicus Sentinel-3 LST data acquired between 2017 and 2021.

**Table 1: Eddy Covariance stations for validating gridded ET maps.**

| Station name | EC site ID | Landcover | Elevation | Data availability* | EBC** |
|---|---|---|---|---|---|
| Puéchabon | FR-Pue | EBF | 270 m | 2000-2021 | 0.82 (-) |
| Lison | IT-Lsn | VIN | 1 m | 2016-2020 | 0.67 (0.71) |
| Muntatschinig meadow | IT-MtM | GRA | 1450 m | 2017-2019 | 0.87 (0.86) |
| Muntatschinig pasture | IT-MtP | GRA | 1550 m | 2017 | 0.80 (0.54) |
| Torgnon | IT-Tor | GRA | 2160 m | 2008-2020 | 1.44 (1.18) |
| Renon | IT-Ren | ENF | 1730 m | 1999-2020 | 1.32 (-) |
| San Rossore 2 | IT-SR2 | ENF | 4 m | 2013-2020 | 0.96 (0.95) |
| Monte Bondone | IT-MBo | GRA | 1550 m | 2003-2020 | 0.96 (1.04)- |

*time span for raw local measurements before excluding years of non-interest
**EBC without brackets corresponds to all available records at the sites, while values in the brackets are derived for the years 2017-2021
EBF = evergreen broadleaf forest, VIN = vineyard, GRA = grassland, ENF = evergreen needleleaf forest

To derive daily ET observations with good quality, all *in-situ* latent heat flux (LE) measurements collected at 30-min

temporal resolution are first pre-processed to eliminate outliers (i.e., records smaller or greater than the 1st and 99th percentile,

respectively), duplicates, rainy events (> 0 mm day$^{-1}$), and eventually days with a sub-daily coverage smaller than 25%

(Hulsman et al., 2023). Additionally, 30-min station records with low-quality assurance (QA) are removed. In case of missing

QA flags, we exclude the corresponding instantaneous data records from further analysis. As a result, the amount of ground

measurements is reduced. Apart from that, the observations are checked for energy balance closure (EBC) ratio (i.e.,

[(H+LE)/(Rn-G)] with Rn: net radiation, G: soil heat flux, H: sensible heat flux), as shown in Table 1. Considering all available

records at the flux sites, EBCs vary between 0.67 at IT-Lsn and 1.44 in IT-Tor, while for our years of interest (2017-2021) the

average ratio ranges from 0.54 at IT-MtP to maximum value of 1.18 at IT-Tor. Even though some stations either exceed unity

(IT-SR2 and IT-Tor) or have smaller values than the acceptable threshold of 0.75 (IT-Lsn, IT-MtP), we include all locations

in the validation process due to small number of flux sites available over the basins (Pastorello et al., 2020). After the quality

checks, the local ET observations are estimated using the approach proposed by *Allen et al.* (1998). Specifically, latent heat

flux [W m$^{-2}$] is converted to daily ET [mm day$^{-1}$] using the following formula:

$$ET \left[\frac{mm}{day}\right] = \frac{LE\left[\frac{W}{m^2}\right] \times 24 \times 60 \times 60 \left[\frac{s}{day}\right] \times 1000 \left[\frac{mm}{m}\right]}{\rho_w\left[\frac{kg}{m^3}\right] \times L\left[\frac{J}{kg}\right]},$$ (1)

where $\rho_w$ is the water density (1000 kg m$^{-3}$), and $L$ represents the latent heat of vaporization (2.25·10$^6$ J kg$^{-1}$). After the equation

transformation, the tower-derived ET is estimated, and then compared against the 100-m ET product as described in the next

sections of this paper.

**2.3 Gridded data**

Multi-source ESA Copernicus data to estimate actual ET are used in this study: satellite, meteorological, and ancillary remotely sensed variables. This chapter outlines source datasets and the accompanying preprocessing steps involved before the main processing chain for deriving ET. Table 2 provides a comprehensive overview of all gridded variables utilized for the ET modeling. It should be mentioned that in Sect. 3 we present more details on all Copernicus-based outputs at intermediate and final processing stages with cloud computing resources.

**Table 2: Gridded data used in this study.**

| Source dataset | Input parameter | Pixel size | Brief summary |
|---|---|---|---|
| Sentinel-3A/B Sea and Land Surface Temperature Radiometer Level 2 | Land surface temperature | 1 km | LST maps at clear-sky conditions based on quality bands provided (ESA, 2022) and downscaled to 100-m spatial resolution |
| Sentinel-2A/B Multispectral Instrument Level 1C and 2A | Surface reflectance (SR) | 10-20 m | Top-Of-Atmosphere and the Bottom-Of-Atmosphere SR bands resampled to 100-m and 1-km pixel size[*] |
| Shuttle Radar Topography Mission (SRTM) | Elevation | 90 m | Digital elevation model (DEM) from the SRTM and its two derivatives: slope and aspect resampled to 100 m and 1 km spatial resolution[*] |
| PROBA-V and Sentinel-3 Ocean and Land Colour Instrument (OLCI) | Landcover | 300 m | Annual maps with global extent derived from PROBA-V (2017-2019) and Sentinel-3 OLCI for the years 2020-2021 resampled to 100 m pixel size |
| European ReAnalysis V5 | Meteorological data | 31 km | Hourly maps of air temperature, vapor pressure, air pressure, wind speed, clear-sky downward solar radiation, and daily all sky downwelling shortwave flux, all matched to Sentinel-3 overpass time |

[*]both 1-km and 100-m datasets are utilized for data-driven thermal downscaling, while 100-m intermediate outputs are incorporated directly into the ET model

In this work, the ET model is forced by Copernicus satellite data, including daily 1-km Sentinel-3 LST maps and fine spatio-temporal resolution Sentinel-2 MSI imagery (10-20-m, 2-5 days revisit time), all derived for the years 2017-2021. Land surface temperature, as a crucial forcing parameter for the ET model, corresponds to daytime S3 acquisitions under clear-sky conditions. Simultaneously, biophysical variables and shortwave bands at 100-m resolution are derived from original S2 at the Bottom-Of-Atmosphere (S2L2A) reflectance maps with a total spatial coverage of 52 Sentinel-2 tiles (Table 2). In case of missing S2L2A, we first pre-process Sentinel-2 Level-1C (S2L1C) to derive atmospherically corrected S2 scenes, as explained in the next sections of this paper. In this work, daytime land surface temperature images, as a crucial forcing parameter for estimating turbulent fluxes, are derived from Sentinel-3A and Sentinel-3B SLSTR data. In this regard, we extract specific bands from 1-km S3 products, including LST, and cloud mask, along with sun and S3 sensor geometries. Due to the two-satellite constellation since June 2018, Sentinel-3 acquisitions with minimum viewing zenith angle (VZA) are selected when multiple scenes on the same day are captured. The reason behind that choice is motivated by the fact that larger VZA has a more negative impact on surface temperature accuracy due to angular anisotropy in the thermal infrared spectrum. To derive surface biophysical variables for ET modeling, the constellation of Sentinel-2 MSI (both A and B) is exploited, and in

particular, 10-20 m nine reflectance bands from the VSWIR region are extracted. In addition, the resulting S2L2A shortwave

channels are used as 100-m predictor variables to downscale 1-km Sentinel-3 SLSTR LST data. More details on satellite data preparation are in the next sections of this paper.

To run the ET processing chain, other two satellite-driven products are also exploited: 300-m Copernicus Climate Change Initiative (CCI) land-cover (LC) maps (*https://www.esa-landcover-cci.org/*) and elevation data obtained from 90-m Shuttle Radar Topography Mission. The first one is derived through the Climate Data Store (CDS) API client in Python as explained

at *https://cds.climate.copernicus.eu/*, while DEM is automatically downloaded from the SRTM-dedicated database available online (*https://srtm.csi.cgiar.org*). After data downloading both inputs are resampled to 100-m resolution to be ready for ET model runs. Despite the pixel size discrepancy between inputs and the ET product, this choice is reasoned by temporal coverage of the CCI LS data (2017-2021) with specially designed look-up tables for estimating ancillary parameters to force TSEB-PT model, such as canopy height, fractional vegetation cover, average leaf size, and canopy shape. In case of elevation, we select

a 90-m SRTM DEM product due to its ET-like spatial resolution. Apart from that, DEM information is used as an input predictor for downscaling 1-km Sentinel-3 LST, and to topographically correct ERA5 parameters following the strategy proposed by *Guzinski et al.* (2021).

Meteorological parameters, which are essential input variables for ET estimation, are derived from high-frequency European ReAnalysis V5 climate data provided by the European Centre for Medium-Range Weather Forecasts (ECMWF) and

downloaded from the CDS for the period 2017-2021 (Hersbach et al., 2020). As forcing inputs for the TSEB-PT model, we use meteorological observations that include air temperature (TA), vapor pressure (VP), wind speed (both $u$ and $v$ components), surface pressure (SP), together with two ERA5 solar radiation components: all-sky shortwave downwelling fluxes and clear-sky downward shortwave radiation ($SW_{in}^{clear-sky}$) temporally matched to Sentinel-3 SLSTR overpass time. Hourly $SW_{in}^{clear-sky}$ datasets are selected rather than all-sky downwelling solar irradiance due to an assumption of clear-sky conditions during S3

acquisitions incapable of penetrating clouds. ERA5 dataset is used for estimating input parameters, like net shortwave radiation and longwave irradiance, and then for deriving instantaneous energy fluxes and extrapolating latent heat flux to daily time steps.

## 3 Methodology

### 3.1 General framework

This study aims to develop an automatic workflow for generating Copernicus-based daily ET datasets at 100-m resolution from 2017 to 2021. Sub-kilometre ET mapping is considered a reasonable scale in the Mediterranean basins characterized by complex topography and highly patched landcover, where 1-km ET maps might not fully represent spatial heterogeneities of the land surface (Massari et al., 2021). In general, the entire workflow to produce a daily 100-m ET product consists of two steps: (1) input parameter preparation, and (2) TSEB-PT modeling of ET (Figure 2). In the following subchapters we describe

each processing chain in more detail.

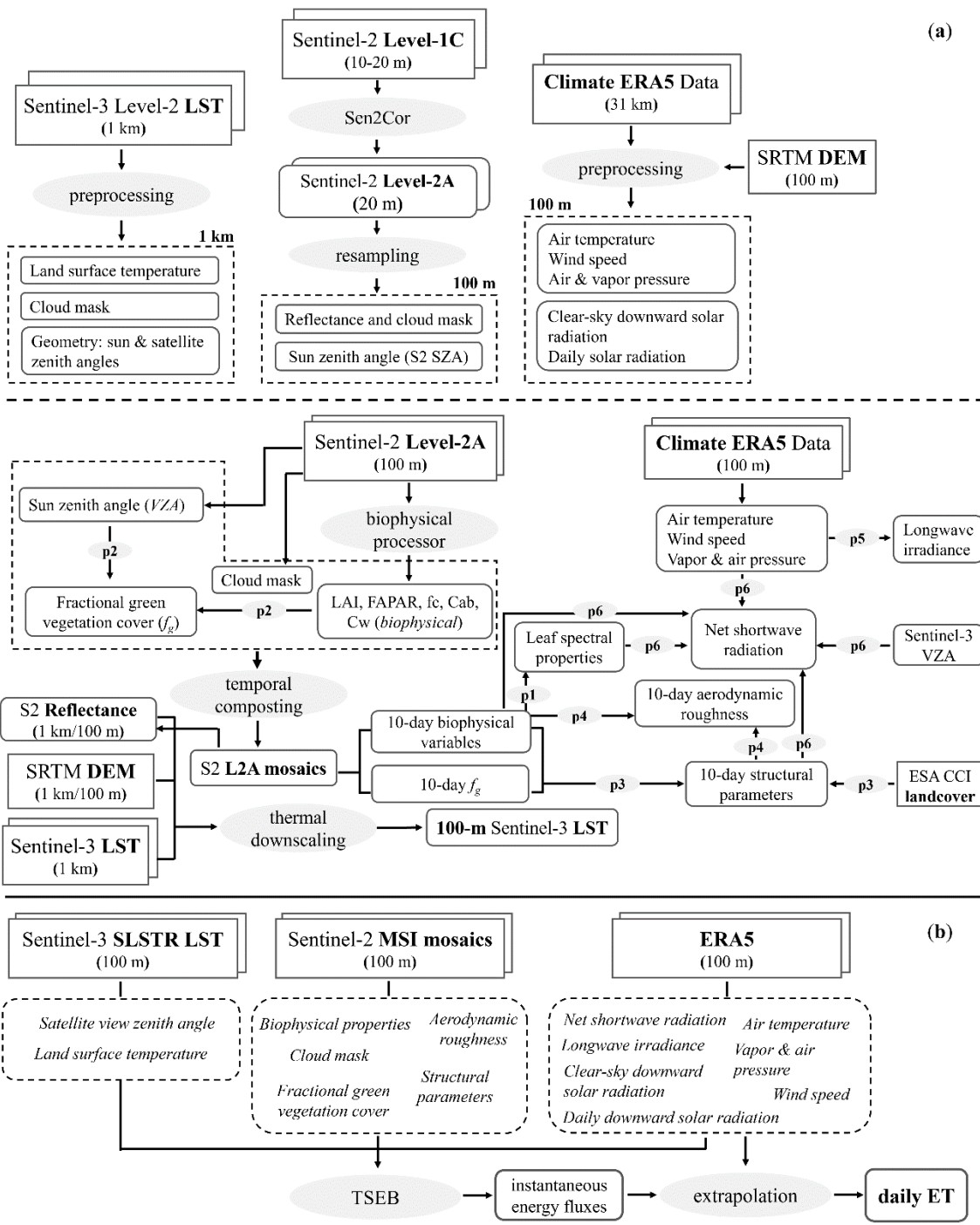

**Figure 2: Schematic flowchart of the entire ET processing including (a) source data preprocessing (the top row) and input retrievals for the Priestley-Taylor Two-Source Energy Balance modeling (the lower line), (b) daily ET estimation at 100 m resolution. While processes are represented by ovals (e.g., preprocessing, thermal downscaling, TSEB-PT modeling, and intermediate p1-p6), rectangles stand for source parameters, and round rectangles indicate intermediate and final results.**

Owing to the large-scale ET modeling and its related high computational and memory requirements, we distribute our work on multiple virtual machines. We implement the processing flow with the setup on Ubuntu v20.04 LTS using two geospatial cloud computing platforms offered by EODC (*https://eodc.eu/*) and ESA High-Performance Computing (HPC) cluster of CloudFerro (*https://cloudferro.com/*). The main advantage of this solution is the direct access to satellite data and the capability to facilitate big data processing more robustly. Apart from these resources, two internal units of EURAC Research with Ubuntu

v18.04 LTS are deployed on VMware machines (*https://www.vmware.com/*). In this case, EO data are directly downloaded from the ESA Copernicus Open Access Hub (*https://scihub.copernicus.eu/*). The multiple selection of cloud providers is motivated by two reasons. First, the entire procedure for deriving ET maps at 100-m resolution is complex (see Figure 2), which translates to big requirements in terms of disk space, computing memory, and processing time. For example, a five-year dataset (2017-2021), including intermediate and final outputs over one Sentinel-2 tile requires around 600 GB. Secondly, we

diversify our processing workflows into many computing units due to their accessibility offered by the ESA Network of Resources (NoR) sponsorship program. Furthermore, in the framework of the 4DMed-Hydrology project EURAC Research was requested to test the new ESA HPC infrastructure considering the high-volume processing of the entire workflow presented in this study.

       To keep the data pipeline consistent over the multiple platforms, we harmonize the entire workflow by creating a unique

Conda environment on all our machines. ET outputs are obtained by automatizing entire routines, including baseline ESA SNAP Graphical Processing Tool (GPT) algorithms and Python baseline codes developed in the framework of Sen-ET project (*https://www.esa-sen4et.org*). For more specific details on the data (pre)processing for both satellite and meteorological parameters, it is recommended to visit the open-source GitHub repository available at *https://github.com/DHI-GRAS/*. The repository includes codes that calculate necessary inputs for the TSEB-PT model.

**3.2 Input parameter preparation**

       Considering multi-source datasets in conjunction with their different processing levels and spatial scales, the primary step of our workflow includes source data preprocessing, and input preparation for forcing the Priestley-Taylor Two-Source Energy Balance model (Table 2, Figure 2a).

       First, we spatially aggregate 20-m Sentinel-2 Level 2A reflectances and their geometries to 100-m pixel size with an

arithmetic mean function. In this study, we select 100-m resolution for the ET product rather than the original 20-m S2 cell size. Prior to final modeling we tested the impact of spatial resolution of input variables on the final ET estimates considering the abovementioned pixel dimensions. ET simulations forced by 20-m and 100-m parameters gave similar results, and thus the latter solution was chosen as a trade-off between high spatial resolution and storage use along with computing speed. In addition, to our best knowledge, long-term high-resolution ancillary variables, e.g., elevation and landcover at spatial

resolution < 100 m are not freely available to be incorporated into the entire workflow.

       Nevertheless, S2L2A preparation is challenging mainly due to incomplete timeseries on the EODC service. This requires copying missing datasets either from CloudFerro or from the Copernicus Open Access Hub. Apart from that, Sentinel-2 data

over Medjerda basin in Africa between 2017 and mid-2018 are not available on all platforms (last accessed on 9 January 2022). As shown in Figure 2a, in that case S2L1C scenes are first pre-processed with Sen2Cor processor to obtain atmospherically

corrected 20-m S2L2A granules together with a scene classification layer for cloud removal afterwards (Main-Knorn et al., 2017). Since the time frame of this work covers the years 2017-2021, it is necessary to use two different versions of Sen2Cor. Indeed, Sen2Cor 2.5.5 is able to ingest only those data belonging to 2017-2021; while S2 scenes from 2022 need to be processed using the Sen2Cor 2.10.01. To solve this problem, we setup two different docker containers for the two Sen2Cor versions. After checking the acquisition date of the input data, the bash script is run considering the time overlapping Sen2Cor

releases. Due to straight-forward Sen2Cor cloud mask retrieval and its well-established workflow, we decided to apply this approach for the entire Sentinel-2 time-series.

Next, S2-driven biophysical parameters, including LAI, canopy height ($h_C$), and fractional vegetation cover ($f_C$), are produced using the S2 Toolbox Biophysical processor (Weiss and Baret, 2016; Xie et al., 2019). The entire procedure requires eight Sentinel-2 L2A bands acquired in the VSWIR electromagnetic spectrum together with geometry information, like sun

and sensor zenith angle. LAI retrieval is a hybrid approach based on an inversion of the PROSAIL radiative transfer model simulations of S2 canopy reflectance using neural network modeling. More details on estimating biophysical variables are provided by *Weiss and Baret* (2016). At the same time, the fraction of green vegetation ($f_g$) is generated by incorporating the S2 VZA and biophysical variables as input parameters (Figure 2b). To minimize the cloud cover effect in Sentinel-2 product, all the abovementioned outputs are mosaicked using a 10-day window with respect to Sentinel-3 overpass dates. This means

that time coincident Sentinel-2 granules are temporally ranked and good quality pixels with possibly the closest date to S3 acquisition are selected. The entire procedure is developed in SNAP GPT to be run in an automated manner over the entire image collection. Next, 10-day composites are utilized in conjunction with other ET model parameters, like 100-m CCI landcover, for deriving remaining inputs, including aerodynamic roughness, along with vegetation structural and spectral properties as shown in Figure 2b. In addition, atmospherically corrected surface reflectance bands are resampled to 1 km spatial

resolution to sharpen daily Sentinel-3 LST data.

As outlined in Sect. 2.3, we use satellite-based land surface temperature from 1-km Sentinel-3 SLSTR Level-2 data for estimating actual ET. Given the sensitivity of TIR instruments to overcast conditions, all cloudy pixels are eliminated with respect to the *cloud-in* mask provided with SL_2_LST product. Next, S3 datasets are cropped according to Sentinel-2 tiles (*https://sentinels.copernicus.eu/web/sentinel/missions/sentinel-2/data-products*) and reprojected to the WGS84 coordinate

system for their synergistic use with Sentinel-2 data afterwards.

Notably, it should be mentioned that CloudFerro provides different versions of Sentinel-3 LST in terms of cropping scheme, baseline collections, and related software versions for data processing. This hinders the immediate use of the data and requires an investment of time to analyze the quality of the data and choose the proper version, and thus S3 LST processing needs to be preceded by data checks and proper filtering. In general, it is recommended that the newest baseline collection (v4) is chosen

rather than a product with baseline collection v3 as the v4 is reprocessed using an upgraded software after major evolutions.

While taking advantage of complementary Sentinel-2 and Sentinel-3 instruments, we sharpen 1-km S3 SL_2_LST product to derive enhanced surface temperatures at a spatial resolution corresponding to our ET product (Figure 2a). Thermal downscaling is based on the Data Mining Sharpener (DMS) approach proposed by *Gao et al.* (2012). It has been successfully applied in many studies to enhance the spatial consistency of LST grids (Anderson et al., 2021; Guzinski et al., 2020; Sánchez et al., 2023; Yang et al., 2021). As presented in *Guzinski and Nieto* (2019), TSEB-PT driven by downscaled DMS-based surface temperatures is more robust compared to ET estimates driven by original 1-km LST data with around a 13% increase in Pearson correlation coefficient (R) between *in-situ* ETs and their corresponding modelled observations. The DMS method incorporated in the Priestley-Taylor Two-Source Energy Balance modeling pipeline has been also demonstrated to be more performant than evapotranspiration estimates derived from METRIC and ESVEP models at 11 flux tower sites across different vegetation types and climate zones (*https://www.esa-sen4et.org/downloads/prototype_evaluation_v1.3.pdf*). On average, TSEB-PT achieved consistently lower Root Mean Square Error and higher correlation for latent flux yielding RMSE of 90 W m$^{-2}$ and R exceeding 0.7, which largely outperforms METRIC and ESVEP by more than 11% and 30% for RMSE and R, respectively. Moreover, *Sánchez et al.* (2023) conducted extensive study on the performance of LST downscaling in Spain, and based on their validation results with *in-situ* measurements the DMS approach gave nearly two times smaller RMSE error compared to the 1-km S3 LST. In addition to the abovementioned literature review, in our co-authored paper we compared Sen-ET outcomes with other evapotranspiration products, including 3-km MSG SEVIRI and 70-m ECOSTRESS ET which on average yielded less robust accuracy metrics than our 100-m retrievals (De Santis et al., 2022). In this regard, the kernel-driven regressions are obtained from the bagging ensemble of decision trees (DT) algorithm that reduces the risk of model overfitting. In this study, DMS predicts land surface temperature at 100 m spatial resolution by exploiting empirical relationships between coarse LST grids and high-resolution explanatory variables for each Sentinel-3 acquisition date. The functional relationship between clear-sky Sentinel-3 LST data and explanatory variables is based on 10-day S2 reflectance composites in conjunction with DEM and shortwave irradiance incident angle at S3 overpass time derived from SRTM-based slope, and aspect grids (Figure 2b). DMS method establishes simultaneously global (within 100-km S2 tile) and local (30 by 30 Sentinel-3 pixels within a moving window) regression models, and then fuses these two estimates as their weighted linear combination to increase the number of samples for model training, and also captures thermal heterogeneity at local scale. Consequently, downscaled LST maps are derived by applying daily models to the HR predictors, while forcing energy conservation between original Sentinel-3 images and sharpened granules. The entire procedure is performed in the blending phase by applying a weight to global and local estimates based on residual correction between the original LST and the 100-m S3 image sharpened with two regression schemes. This means that LST pixels with lower bias result in bigger weight, while grid cells with larger residuals have a smaller impact on the final LST estimation.

Given a wide range of fine resolution predictors (< 100 m) and their high revisit time (2-5 days), the enhancement method in the spatial domain is selected rather than the image fusion approach that increases temporal availability of high spatial resolution LST images by exploiting (sub)daily observations from coarse resolution TIR scanners (Sun et al., 2017; Yang et al., 2017). As mentioned before, the DMS method belongs to well-established downscaling approaches and its recent open-

source implementation increases its visibility among users (Guzinski and Nieto, 2019). Furthermore, the TSEB-PT has been constantly updated to enhance the modeling strategy for thermal sharpening, and as reported by *Guzinski et al.* (2023), the enhanced DMS achieved better results translating into up to 1.5 K improvements in accuracy of downscaled LST and average RMSE of 0.8 mm day$^{-1}$ for daily ET. The successful applications of the DMS procedure for deriving high spatial resolution ET, as shown in many research studies before, moved us towards generation of 100-m ET dataset assuming its better performance in different land covers and climates compared to original S3-driven TSEB-PT estimates at 1 km resolution. Nevertheless, considering terrain complexity and patched landcover of our study areas compared to the abovementioned studies where majority of flux towers are located in relatively homogenous environments (Table 1), we additionally evaluate the performance of the ET model forced by original 1-km Sentinel-3 SLSTR versus ET forced by 100-m DMS-derived inputs.

ECMWF ERA5 climate datasets also require preprocessing prior to inclusion in the Two-Source Energy Balance model. All extracted variables from the reanalysis dataset (see Sect. 2.3 for more details), except for wind speed, are corrected for terrain effects using the SRTM DEM product (Figure 2a). Similar to instantaneous variables, all-sky shortwave downwelling fluxes are first enhanced by accounting for topography orientation (i.e., illumination conditions) and an hourly cloud cover factor derived from SW$_{in}$$^{clear-sky}$, and then interpolated to daily observations. Considering better representativeness of low-resolution meteorological parameters at the blending height of 100 m rather than 2 m above ground, TA, VP along with SP are additionally recalculated at that height (Guzinski et al., 2021). Daily average solar radiation is obtained by interpolating hourly ERA5 shortwave downward irradiance over a 24-hour period starting at midnight local time. Next, the product is used to extrapolate instantaneous latent flux to daily ET. After calibrating ERA5 components, we prepare specific radiative fluxes for deriving instantaneous energy fluxes corresponding to Sentinel-3 overpass. In this regard, meteorological input is utilized to compute longwave irradiance, and then instantaneous net shortwave radiation is derived from Sentinel-3 VZA imagery and 10-day Sentinel-2 composites of structural and biophysical parameters (Figure 2a).

### 3.3 TSEB-PT modeling of ET

In this work, we utilize the Priestley-Taylor Two-Source Energy Balance model driven by ESA Copernicus data (Figure 2) to produce daily evaporation maps over the Mediterranean region (Norman et al. 1995, Kustas and Norman, 1999). The main advantage of the model over heterogenous areas is the fact that TSEB-PT considers the soil (S) and canopy (C) as two distinct components and employs a two-layer approach to estimate latent (LE) and sensible heat (H) fluxes for each element separately:

$$Rn_S = LE_S + H_S + G, \tag{2}$$

$$Rn_C = LE_C + H_C, \tag{3}$$

where Rn denotes the net radiation [W m$^{-2}$], LE represents the latent heat flux [W m$^{-2}$], H represents the sensible heat flux [W m$^{-2}$], and G stands for soil heat flux [W m$^{-2}$]. Unlike other satellite-based methods, the model minimizes the number of input parameters, and its relative simplicity makes it an ideal candidate for high-volume processing (Kustas and Anderson, 2009).

The net radiation sub-components (Rn$_S$ and Rn$_C$) are calculated following the methodology presented by Campbell and Norman in 1998. The H$_S$ (H$_C$) is determined by evaluating the temperature gradient between the soil (canopy) and the air temperature (TA) at a reference height, as described by *Guzinski et al.* (2020). The primary remotely sensed variables required by the model are land surface temperature, which represents the combined effect of both soil and canopy, and fractional vegetation cover (f$_C$), used for partitioning the energy between vegetation cover and soil. As the surface temperatures of soil (LST$_S$) and vegetation (LST$_C$) are unknown, TSEB-PT divides LST into soil and canopy temperatures based on the fractional vegetation content, which is parametrized by leaf area index (Guzinski et al., 2014):

$$LST = (f_C LST_C^4 + (1 - f_C)LST_S^4)^{0.25}, \tag{4}$$

TSEB-PT employs an iterative procedure to calculate LST$_S$ and LST$_C$, along with their respective soil (canopy) sensible heat fluxes H$_S$ (H$_C$). The entire process of determining LST$_S$, H$_S$, LST$_C$, and H$_C$ commences with an initial estimation of canopy transpiration (LE$_C^{int}$) based on the Priestley-Taylor coefficient $\alpha_{PT}$ (Priestley and Taylor, 1972):

$$LE_C^{int} = \alpha_{PT} f_g Rn_C \frac{\Delta}{\Delta + \gamma}, \tag{5}$$

where $\Delta$ is the slope of the vapor pressure versus air temperature, and $\gamma$ is the psychometric constant [kPa K$^{-1}$]. Having initial LE$_C^{int}$, the sensible heat flux from vegetation is calculated as the residual term of the energy balance (Equation 3). Consequently, LST$_C$ is obtained from the estimated H$_C$ and air temperature as explained in *Nieto et al.* (2019). Next, LST$_S$ is obtained using Equation (4), and subsequently, the soil sensible heat flux is derived. Finally, LE$_S$ is determined as the residual flux from Equation (2), which ensures energy balance closure. The resistance term is formulated following the approach proposed by *Kustas and Norman* in 1999. The entire iterative process to derive turbulent fluxes terminates when the soil latent heat flux reaches a non-negative value. If the obtained LE$_C^{int}$ does not yield a physically realistic solution, the $\alpha_{PT}$ (with an initial value of 1.26) is successively modified by decreasing it until a physically realistic solution is obtained (Norman et al., 1995; Kustas and Norman, 1999). More details on TSEB-PT may be found in many research studies (Chintala et al., 2022; Guzinski et al., 2020; Hoffmann et al., 2016; Nieto et al., 2019) and open-source Github repository written in Python at *https://github.com/hectornieto/pyTSEB*.

In this study, the TSEB-PT model is forced with gridded inputs, including Sentinel products in conjunction with ERA5 climate parameters for deriving 100-m instantaneous energy fluxes corresponding to Sentinel-3 overpass. As shown in Figure 2b, radiometric temperature and its viewing zenith angle are sourced from the S3 LST product, while biophysical input parameters, like chlorophyll- and water-based leaf reflectance and transmittance, f$_C$ and its green component f$_g$, are derived from 10-day Sentinel-2 mosaics (see Sect. 3.2 for more detail). While vegetation structural parameters, including leaf angle distribution together with canopy width-to-height ratio variables, are used to compute the clumping index for quantifying foliage distribution, aerodynamic roughness length and zero-plane displacement height serve as inputs for parametrizing resistances required by the TSEP-PT. As summarized in Figure 2b, apart from S3 LST and S2 SR-driven variables, hourly ERA5 derivatives are utilized. In this regard, meteorological observations, which include air temperature, wind speed, vapor,

and air pressure, are interpolated on a 30-min time scale fitted to Sentinel-3 acquisitions, and then applied to the model. Furthermore, longwave irradiance and net shortwave radiation are incorporated into TSEB-PT for estimating the energy exchange between the surface and the atmosphere. After energy fluxes are obtained (e.g., LE, H, Rn, and G), total latent heat flux is extrapolated to daily ET estimates over each S2 tile using the all-sky ERA5 daily solar irradiance. The last step of the processing flow involves the generation of daily ET maps for our subdomains that include Po, Ebro, Hérault, and Medjerda basins. This is achieved by average mosaicking spatiotemporally tiles using a specially developed compositing algorithm as described in Sect. 4 of this paper.

### 3.4 ET validation

To access the quality of daily 100-m ET product, we validate our results using local measurements collected by Eddy Covariance systems (Pastorello et al., 2020) and also compare the model performance with ET retrievals forced by 1-km Sentinel-3 LST. In this regard, we perform ground-based validation by exploiting relationships between *in-situ* daily latent heat fluxes and ET estimates derived from TSEB-PT. As mentioned in Sect. 2.2, before the validation, LE values are converted to ET estimates expressed in millimetres per day (Allen et al., 1998), and then the resulting datasets are spatiotemporally matched to the 5-year ET dataset (2017-2021). Validation of gridded TSEB-PT outputs requires information on the spatial range of EC towers. Due to parameter-demanding methods for estimating two-dimensional flux footprints, such as the Flux Footprint Prediction climatology proposed by *Kljun et al.* (2015), two simplified validation strategies are chosen: pixelwise and buffer strategies within 100-m cell grid and 50-m radius, respectively. While the first approach is based on the direct extraction of pixel values to points, for the latter method satellite-based TSEB-PT simulations are derived within a 50-m extent around each EC site considering the percentage contribution of each overlapping 100-m pixel in that zone. TSEB-PT validation is performed when ET data cover at least 50% of the total buffer area. In this work, the discrepancies between modelled and observed ET values are evaluated by means of statistical accuracy metrics that include the Root Mean Square Error (RMSE), Pearson correlation coefficient, and mean bias (MB). They are calculated as follows:

$$RMSE = \sqrt{\frac{\sum_{i=1}^{n}(x_i - y_i)^2}{n}}; \quad R = \frac{\sum_{i=1}^{n}(x_i - x_{mean})(y_i - y_{mean})}{\sqrt{\sum_{i=1}^{n}(x_i - x_{mean})^2 \sum_{i=1}^{n}(y_i - y_{mean})^2}}; \quad MB = \frac{\sum_{i=1}^{n}(y_i - x_i)}{n}, \quad (6)$$

where $x_i$ stands for the ground-based value on the day $i$, $y_i$ denotes the predicted value from the TSEB-PT model for a daily observation $i$, and n is the number of matching observations incorporated in the validation process.

### 4 Results and discussion

#### 4.1 Evaluation of daily ET at the EC sites

The performance of the ET maps is evaluated against *in-situ* ET data derived from the EFDC database over grasslands (IT-MBo, IT-MtM, IT-MtP, IT-Tor), forest (FR-Pue, IT-Ren, IT-SR2), and vineyard at Lison site (IT-Lsn). First, we examine

overall relationships between local ET and TSEB-PT based estimates under clear-sky conditions together for all sites (Figure 3a). As illustrated in the global scatterplot, the linear regression analysis over all the sites with pixelwise (buffer) approach generates on average RMSE and R of 1.38 mm day$^{-1}$ (1.39 mm day$^{-1}$) and 0.60 (0.59), respectively. Due to insignificant differences between the two validation strategies applied, in the further part of this study, we focus on the point-based approach. Note that even though half of the EC towers cover the 4 years between 2017 and 2020 as shown in Table 2, data filtering and its related quality checks together with the necessity to match *in-situ* points with gridded ET values result in a lower number of paired observations for the validation (Table 2, Figure 3).

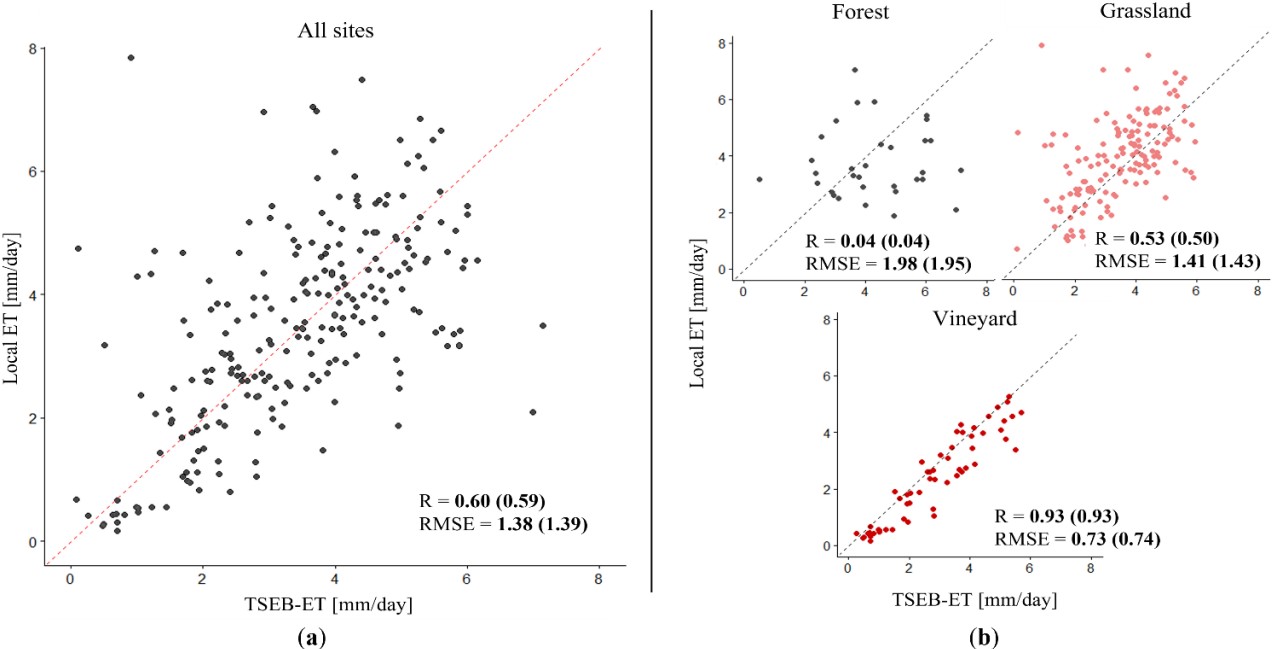

(a)                                                                                (b)

**Figure 3: Scatterplots between ground-derived ET observations and TSEB-PT simulations for: (a) all Eddy Covariance sites included, (b) separated landcovers: forest grassland, and vineyard. Note that values in brackets indicate accuracy statistics obtained within 50-m buffer, while records without parentheses represent pixelwise validation strategy.**

Second, we investigate TSEB-PT performance considering separately three vegetation biomes, i.e., forest, grasslands, and vineyards. As shown in Figure 3b, the accuracies differ between the landcovers with the best results for lowland vineyards in IT-Lsn yielding RMSE of 0.73 mm per day, and a high score for Pearson coefficient exceeding 0.90. Meanwhile, for grasslands, we observe a moderate correlation between *in-situ* and modelled ET with R = 0.53 and a larger mean error of around 1.41 mm day$^{-1}$ (Figure 3b). Notably, it should be mentioned that all grass-covered EC locations are situated over the Alpine region at altitudes ranging from 1450 to 2160 m a.s.l characterized by relatively steep slopes and high landcover variability (see Appendix A). These factors together with low-resolution inputs like downward and net solar radiation might affect the TSEB-PT performance at the Muntatschinig (IT-MtM and IT-MtP) and IT-Tor sites (Table 3). On the other hand, despite the alpine location of IT-MBo, the TSEB-PT exhibits a larger potential for estimating daily ET as presented in Table 3. The evaluation metrics for this station indicate a slight underestimation of the model (MB = -0.16 mm day$^{-1}$) yielding RMSE

and R of around 1 mm day$^{-1}$ and 0.7, respectively. The better accuracy statistics over IT-MBo compared to other grass-covered sites are attributable to the relatively homogenous land cover and flat terrain at IT-MBo. The distribution of solar radiation,

wind speed, and air temperature gradients are less influenced by a landscape complexity over mountain plateau than over steep slopes, and thus coarse resolution ERA5 might be more representative for IT-MBo compared to IT-Tor and Muntatschinig locations.

**Table 3: Local accuracy scores between *in-situ* data and modelled ET using TSEB-PT for the eight Eddy Covariance towers incorporated in this study.**

| Site name | Landcover | R | RMSE | MB |
|---|---|---|---|---|
| IT-Lsn | vineyard | 0.93 | 0.73 | 0.45 |
| IT-MBo | | 0.71 | 1.02 | -0.16 |
| IT-MtM | grassland | 0.39 | 2.59 | -1.47 |
| IT-MtP | | 0.74 | 2.12 | 2.06 |
| IT-Tor | | 0.38 | 1.89 | -1.51 |
| FR-Pue | | -0.18 | 2.06 | 0.82 |
| IT-Ren | forest | 0.62 | 2.10 | -1.72 |
| IT-SR2 | | 0.23 | 1.86 | 1.18 |

The average accuracy scores over forested areas are the least satisfactory when predicting daily gridded ET. The Pearson correlation coefficient is negligible (R = 0.04), and the RMSE error is around 2 mm day$^{-1}$ (Figure 3b). Moreover, as seen in Table 3, regardless of the site and forest type (evergreen broadleaf or needleleaf forest), the TSEB-PT based ET maps have high RMSE scores ranging from 1.86 mm to 2.10 mm per day with big over(under)estimation of 1.18 mm day$^{-1}$ (-1.72 mm day$^{-1}$) for IT-SR2 and IT-Ren, respectively. The poor accuracy at forested sites might be related to the possible EC

measurement uncertainties associated with surface energy imbalance (see Table 1) and coordinate rotation of turbulent fluxes. These aspects shall be further investigated by removing unsatisfactory in-situ observations through applying stricter criteria for energy imbalance, and also by exploiting different methods for calculating coordinate system for flux retrieval at challenging EC towers (Castelli et al., 2018; Mauder et al., 2013; Rannik et al., 2020; Ross and Grant, 2015). Furthermore, the robustness of the TSEB-PT is also affected by land surface features, such as complex tree structures and their multilayer

composition which is not considered in Sen-ET. This means that controlling parameters for the model obtained from remote sensing (Copernicus Sentinels) and climate reanalysis data (ERA5) may not capture spatial variability of vegetation elements often accompanied with shadows (Penot and Merlin, 2023). Similar to our results, in Jaafar et al. (2022) TSEB-PT forced by LST derived from Landsat and MODIS was found less robust over MF and EBF forests (i.e., RMSE of 1.5-3 mm day$^{-1}$) compared to its superior performance over croplands with a mean error below 1.4 mm per day. Even though accuracy scores

indicate a close agreement with our findings over forested landscapes, the authors obtained higher correlation with a minimum R of 0.6 in woody savanna. Therefore, further work will concentrate on increasing TSEB robustness over complex landscapes by enhancing the model with better quality input variables (e.g. landcover, canopy height, solar radiation and wind speed), together with adjusting default value of the $\alpha_{PT}$ coefficient that depends on climate and vegetation biophysical properties

(Andreu et al., 2018; Cristóbal et al., 2020; Guzinski et al., 2013). Furthermore, Yang et al. (2017, 2020) applied ALEXI/DisALEXI using multi-sensor TIR data fusion approach (e.g., GOES, MODIS, and Landsat) to derive 30-m daily ET retrievals at pine forest sites, and showed a good correspondence against flux towers with an average RMSE ranging from 1.0 mm day$^{-1}$ to 1.3 mm day$^{-1}$. Nevertheless, the authors suggest Landsat-based modeling for deriving high spatial resolution ET rather than medium scale MODIS LST, especially over heterogenous forested landscape to account for complex structure of these biomes.

In parallel, we investigate the impact of 1-km Sentinel-3 LST on final accuracy of ET product. In this regard, TSEB-PT is rerun with low resolution surface temperatures, and obtained outputs are compared against in-situ daily ET that temporally overlap with 100-m gridded retrievals. As shown in Table 4, on average, the comparison results between two gridded products demonstrates better TSEB-PT prediction skill with downscaled temperature rather than with original Sentinel-3 LST product. In general, we observe a 13% decrease in RMSE and an improvement in Pearson correlation coefficient by around 12%, which shows a closer agreement between ground measurements and 100-m ET retrievals when all flux sites are incorporated in the analysis. At the level of a single landcover class ET model driven by downscaled LSTs outperforms TSEB-PT estimates derived from 1-km Sentinel-3 data with the most satisfactory accuracy results for grassland and vineyard. For these biomes RMSE (R) values range between 0.73 mm day$^{-1}$ (0.94) and 1.44 mm day$^{-1}$ (0.49) which corresponds to a 15% improvement in accuracy metrics compared to 1-km ET retrievals (Table 4). In the case of forested sites, the validation scores are very similar for both ET products with no enhancement observed for high resolution outputs. More detailed information on accuracy scores at flux site level is provided in Appendix B.

**Table 4: Station-based comparison of accuracy results between 1-km and 100-m ET retrievals considering RMSE errors and R scores.**

|  | RMSE | | R | |
| --- | --- | --- | --- | --- |
|  | 1 km | 100 m | 1 km | 100 m |
| **All sites** | 1.62 | 1.41 | 0.49 | 0.55 |
| **Grassland** | 1.73 | 1.44 | 0.40 | 0.49 |
| **Forest** | 2.10 | 2.18 | -0.12 | -0.11 |
| **Vineyard** | 0.84 | 0.73 | 0.87 | 0.94 |

To sum up, the evaluation metrics depict a high landcover dependency with the best accuracy for the vineyard site in IT-Lsn and the grass-covered plateau of IT-MBo yielding an average RMSE ranging from 0.7 mm day$^{-1}$ to 1 mm day$^{-1}$ and a mean R coefficient of 0.7-0.9. Despite 100-m Sentinel-based inputs, it is still challenging to accurately estimate ET over mountains and forest areas, like alpine grasslands (IT-MtM, IT-MtP, and IT-Tor) and Mediterranean forests in FR-Pue and It-SR2. For these ecosystems mean statistics scores are less satisfactory with RMSEs ranging from 1.86 mm day$^{-1}$ at IT-SR2 site to 2.59 mm day$^{-1}$ at IT-MtM (Table 3). This might appear due to several reasons. First, these sites are characterized by patched land covers, including grazed grass, sparsely distributed bushes, and trees, along with their complex multilayer structure exposed to shadows, which contributes to spatial heterogeneities within 100-m ET pixels. Furthermore, high-elevation grasslands are

strongly affected by complex interactions between surface energy balance components having an impact on final ET estimates (Mildrexler, 2011). These findings are in line with the outcomes of *Bartkowiak et al.* (2022) where MODIS LST-based TSEB-PT achieved a moderate agreement against local LE records at Muntatschinig sites ($R^2$ close to 0.61), even though the model was forced with ground-derived meteorological inputs at very high spatiotemporal resolution.

Owing the temporal lag of vegetation response between VSWIR and TIR spectrum, reflectance bands from Sentinel-2 MSI might be insufficient to estimate LST, particularly over non-homogenous areas with patched landcover and complex topography. As reported in other studies, ancillary parameters like soil moisture, emissivity, and other surface energy balance components are expected to be more robust explanatory variables to predict surface temperatures at higher spatial resolution (Hu et al., 2023; Merlin et al., 2010). Furthermore, the 10-day reflectance composites obtained from Sentinel-2 L2A may not be temporally representative of vegetation conditions corresponding to Sentinel-3 acquisition dates introducing additional uncertainties in DMS procedure, especially for areas with fast canopy changes due to harvesting, livestock grazing, and mowing events. Consequently, this also affects downscaled LST images that are predicted from the S2 SR mosaics neglecting time-varying characteristics of the land surface.

Apart from the lagged S2 response, the data driven DMS sharpener algorithm depends on thermal variability of 1-km S3 pixels, which constrains its predictions within those cell values used for model training (see Sect. 3.2 for more details). This could contribute to bigger differences between *in-situ* and modelled ET.

As mentioned before, we use the existing TSEB-PT configuration with the global ERA5 data to scale up the ET retrieval over the Mediterranean region. The ET models are controlled by climate inputs derived from 31-km fields, which might negatively influence energy fluxes and daily ET. As reported by *Fisher et al.* (2017), these data might not capture local climate conditions and their rapid changeability over complex areas. By applying 31-km meteorological inputs together with net and downward shortwave radiation, ERA5 parameters do not reflect spatial variability of the land resulting in a mismatch between EC tower-derived ET values and their corresponding gridded estimates.

## 4.2 Daily ET mosaics

The last step of the processing workflow includes the generation of daily ET composites from S2 tile-cropped ET grids for each basin separately. In this regard, we create daily mosaics with respect to acquisition days of the SL_2_LST products considering two different product dissemination units (PDUs) that include Sentinel-3 images distributed in half-orbit stripes and 3-min frame mode. Given many ET maps with the same date within an identical Sentinel-2 tile, which can happen to the frame scenes with smaller spatial extents, ET granules are first combined by applying an average function over overlapping areas for each product derived from individual S3 SLSTR instruments. Next, day-coincident ET grids obtained from Sentinel-3A and Sentinel-3B satellites are together composited. In the case of single frame acquisitions and large stripe images that processing step is skipped. Finally, all ET granules are fused concerning the S2 tile-wise basin coverage (see Appendix C) where overlaying areas of different S2 tiles are also averaged. Low-quality pixels, mostly affected by clouds and lack of the input parameters for the TSEB-PT, are flagged as NaN values. To make the datasets harmonized and consistent with other

hydrological products all mosaics are reprojected to the WGS84 coordinate system using a uniform MR domain grid (Massari et al., 2022).

The development of the ET mosaics compositing scheme has been performed through the SNAP Graphical Processing Tool. Figure 4 presents the final ET maps in the summer of 2018 for the MR catchments that include Ebro, Hérault, Medjerda, and Po basins.

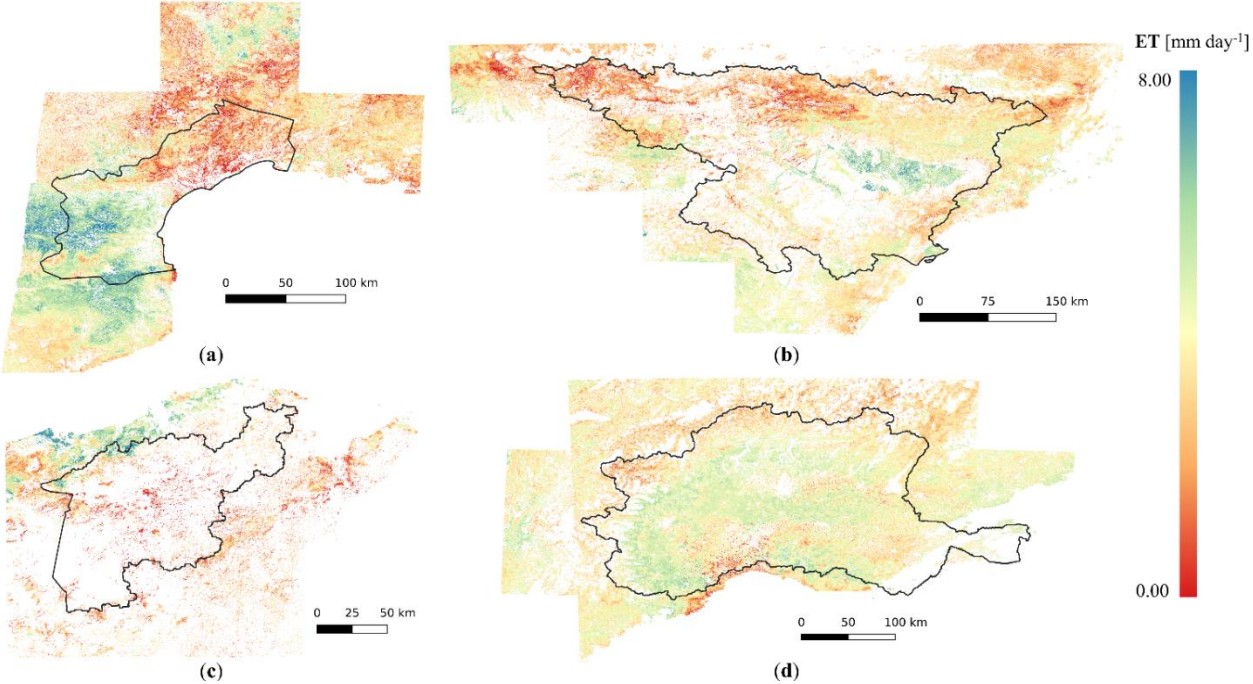

**Figure 4: Examples of 100-m ET mosaics from the third week of August 2018 for (a) Languedoc-Roussillon with Hérault (15.08), (b) Ebro (13.08), (c) Medjerda (16.08), and (d) Po (15.08) basins. The entire grids are cropped following the Sentinel-2 tiling scheme. Missing data (in white) are related to Sentinel-3 cloud masks and anthropogenic surfaces. Since we focus on terrestrial ET, water bodies are also removed.**

From a visual assessment of the daily mosaics, regardless of study areas, no notable irregularities in evaporation are
observed (Figure 4). Daily ET composites follow both landscape and season-induced ET patterns across the study areas. As shown in Figure 4, between the 13th and 15th of August, the Ebro and Po basins contain lower ET values over mountain regions of the northern Apennines and the southern Alps between Italy and Switzerland, along with the Pyrenees and Iberian Range in Spain (Figure 4b, d). At the same time, the central parts of the Ebro catchment, despite semi-arid climatic conditions, receive higher ET values, which indicates a strong impact on agricultural activities where many parcels are irrigated during the growing
season. Similarly, more intense ET can be observed in the western Alps and across the Po river depression covered with extensive canal systems allocated to farming and food production. In parallel, Hérault exhibited two different ET zones on the 15th of August 2018 (Figure 4a). While the southwest part of the Pyrenees is characterized by greater daily ET rates, the remaining area represented by arable land indicates generally lower evaporation proving possible water stress in the LR region characterized by hot and dry summers. The daily composite of the Medjerda basin from August 16th, despite many invalid

pixels in the image, depicts geographically reasonable TSEB-PT model estimates (Figure 4c). Namely, the seaside area in the north represents irrigated fields where ET is expected to be larger compared to the southern zone with limited water resources due to the arid climate.

Nevertheless, the ET product contains areas where TSEB-PT is incapable of estimating ET. Apart from masked surfaces like water bodies and other non-vegetated classes, we can observe an impact of cloud cover (CC) in the final ET product. Even
though we reduce this effect by combining decadal Sentinel-2 composites, which minimize cloud probability, and daily Sentinel-3 LST that by its high revisit time enables relatively continuous ET monitoring without big temporal gaps between observations, cloud occurrence is still present in the daily ET mosaics. This is especially visible over the Medjerda catchment where nearly 86% of the total vegetated surface area contains non-valid pixels (Figure 5a). Seaside areas and the central part of the MB might be affected by overcast conditions, while the southern part is covered by desert and sparse vegetation. In
addition, smaller number of S2 acquisitions outside Europe might have an impact on higher CC percentage in Tunisia. On the other side, the three remaining basins (i.e., EB, HB, and PB) are generally influenced by relatively similar frequencies of overcast conditions that range between 62% for Hérault and 69% for Po and Ebro as shown in Figure 5a. Similar to Medjerda region, they are located in the close range of Mediterranean Sea, and in case of Ebro and Po they include extent mountain ranges, like the Alps and Pyrenees, frequently exposed to cloud contamination.

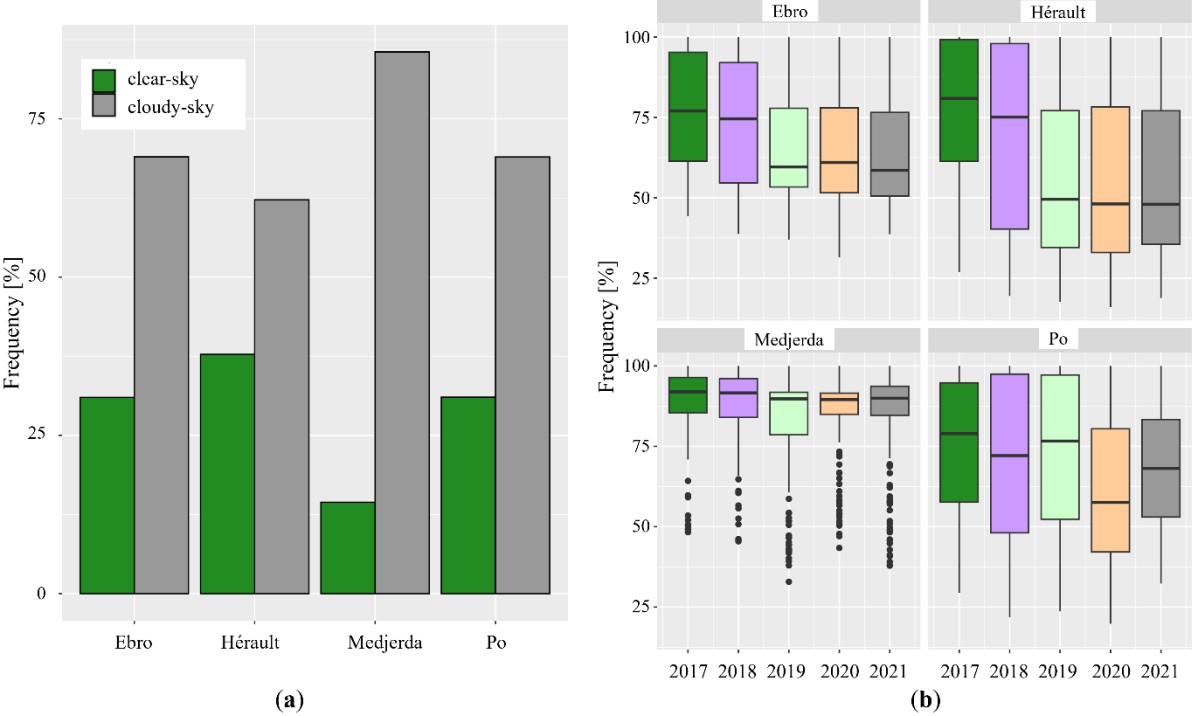

(a)                                                                                (b)

**Figure 5: General overview on the sky conditions for the final ET mosaics over each basin (EB, HB, MB, PB) considering: (a) clear and overcast conditions at Sentinel-3 and Sentinel-2 overpass time for the period 2017-2021, (b) the distribution of cloudy-sky conditions for each separate year. Note that non-vegetated surfaces are excluded from this analysis. In order to minimize other disturbing factors such as a seasonal snow cover, our time of interest covers the period during growing season.**

Nonetheless, if we scrutinize the cloud occurrence over the entire ET collection at annual basis (2017-2021) the frequency of cloudy events fluctuates from day to day (Figure 5b). For example, considering the geographical proximity of Ebro and Hérault basins, we can observe higher values of cloud coverage within the period 2017-2018 with an average CC ($CC_{mean}$) equal to 74%, while for 2019-2021 clear sky pixels correspond to 40% of all available grid cells. At the same time, the interquartile ranges (IQRs) are relatively consistent over time for each of two catchments. On average, cloud cover frequency over Po basin differs between years with the biggest IQRs values in 2017-2019 ($CC_{mean}$ = 72%), and smaller cloud effect yielding 64% for 2-year period between 2020 and 2021 (Figure 5b). In contrast, Medjerda region is the most affected by clouds with the highest cloud coverage and lowest CC variability among all basins over the entire time period. As depicted in Figure 5b, an average percentage of invalid pixels over the catchment ranges from 83% in 2019 to 88% in 2017. In general, sky conditions with more than half of clear-sky pixels are observed beyond the IQRs.

## 5 Code and data availability

The 100-m ET maps are the contribution of EURAC Research to the 4DMED-Hydrology project funded by the European Space Agency (*https://www.4dmed-hydrology.org/*). This dataset is available for the period 2017-2021 for each separate Mediterranean basin that includes Ebro, Hérault, Medjerda, and Po catchments. ET maps are produced for each month (Jan-Dec) of the year in the form of daily observations. The spatial extent corresponds to Sentinel-2 tiling grids overlapping with the study domains (Figure 4). Each layer contains one single band with daily ET values [mm/day] corresponding to Sentinel-3 acquisition day. Invalid pixels, mainly due to vegetation masks, cloud contamination, and lack of input data for the TSEB-PT model, are filled with NaN values. ET outputs are generated in Cloud Optimized GeoTIFF (COG) format with metadata included in the file attributes. A COG is a regular GeoTIFF file, optimized for use in a cloud environment, ready to be hosted on a HTTP(S) file server, with an internal organization that enables more efficient workflows on the cloud. It facilitates data download through HTTP(S) data request and allows a user to crop data according to area of interests. The importance of metadata availability in the COG file is twofold: on one side it offers the chance to evaluate the data without opening it; on the other side, this information has been used to fill a catalogue following the SpatioTemporal Asset Catalog (STAC) specifications that provides a common structure for describing and cataloguing spatiotemporal assets (*https://stacspec.org/en/*). To query the ET data in the STAC catalogue, a relevant STAC documentation is available together with a Python snippet code for massive data download via the Environmental Data Platform (*https://edp-portal.eurac.edu/discovery/*). Alternatively, we provide an additional data link to manually download the entire basin-based ET collection.

As mentioned before, daily ET datasets are accessible through the Environmental Data Platform of EURAC Research under the Creative Commons Attribution 4.0 License (CC BY 4.0). Note that if you use these datasets, you are kindly asked to include the following references concerning the four study domains:

(1) For Ebro: https://doi.org/10.48784/b90a02d6-5d13-4acd-b11c-99a0d381ca9a (Bartkowiak et al., 2023a),

(2) For Hérault: https://doi.org/10.48784/fb631817-189f-4b57-af6a-38cef217bad3 (Bartkowiak et al., 2023b),

(3) For Medjerda: https://doi.org/10.48784/70cd192c-0d46-4811-ad1d-51a09734a2e9 (Bartkowiak et al., 2023c),

(4) For Po: https://doi.org/10.48784/7abdbd94-ddfe-48df-ab09-341ad2f52e47 (Bartkowiak et al., 2023d).

All code routines developed for the entire processing workflow are available upon request from the authors.

## 6 Conclusions

Although ET plays a key role in the hydrological cycle and represents a nexus between energy, water, and carbon exchange, its availability is constrained either to short-range *in-situ* measurements or freely available satellite-derived ET data at coarse spatial resolutions ($\geq$ 0.5 km). Thus, the generation of HR ET datasets is of high importance among scientists, but also of governmental institutions and agricultural communities to advance hydrological cycle monitoring and sustainable water management.

Motivated by the lack of ET data at a high spatiotemporal resolution over the Mediterranean region, we build an automatic workflow to provide 5-year time-series of 100-m daily ET product (2017-2021) as a helpful tool in the context of recurring drought events across four MR basins: Ebro (Spain), Hérault (France), Medjerda (Tunisia), and Po (Italy). Specifically, we utilize a globally applicable TSEB-PT approach with a reduced number of inputs that minimize its complexity, being an advantage for high-volume processing at large scales. The model is physically based and has a long history of successful 645   research studies that confirm its maturity and stability. Results demonstrate that the developed TSEB-PT ET workflow is capable of predicting ET in a robust manner.

The daily composites generally follow the seasonal patterns of the canopy over the study areas (see Sect. 2.1) with higher ET values over irrigated areas and lower estimates over rainfed vegetation like natural grasslands and forests. The validated 100-m time-series with local records from Eddy Covariance stations have more reasonable scores compared to time-coincident 650   ET estimates forced by 1-km LST data resulting in lower RMSE and higher R scores. Nevertheless, the prediction skill of the Copernicus-driven ET from TSEB-PT exhibits landcover dependency with the best accuracy results for agricultural areas and less satisfactory outcomes in forests. The validation scores confirm a strong agreement between gridded data and *in-situ* measurements, especially over relatively uniform and flat areas represented by vineyard and grassland yielding an average R and mean error equal to 0.80 and 0.85 mm day$^{-1}$, respectively. In contrast, on a level of single landcover classes, RMSEs (Rs) 655   range from 1.86 mm day$^{-1}$ (-0.18) in forested areas to 2.59 mm day$^{-1}$ (0.74) for grasslands considering all remaining sites in this study.

Even though the Priestley-Taylor TSEB-PT gives very promising outcomes for plain and homogenous areas, which makes it a perfect candidate for lowland agriculture activities, site locations across biomes, like steep mountain grasslands and forested sites, create some confusion to the model mainly due to inability of gridded inputs to represent complex canopy structure and 660   its heterogeneity together with highly changing meteorological conditions (Elfarkh et al., 2020). Notably, TSEB-PT estimates are affected by spatial heterogeneities of the study areas, and consequently pixel size of gridded input parameters. Given the temporal and spatial frames of this project, the validation is conducted against EC towers, 62.5% of which are located in the

Alps. Thus, future work should be extended to more validation sites with relatively simple terrain and homogenous vegetation to minimize environmental impacts on the TSEB-PT performance, and also in case of complex sites EC processing procedures shall include more strict quality checks and enhancement procedures by exploiting capabilities of different energy balance closure methods, and correcting EC coordinate system for *in-situ* flux retrieval. In parallel, additional work shall be focused on the implementation of better-quality inputs such as climate forcings from 3-km solar radiation acquired by MSG SEVIRI and 5.5 km CERRA reanalysis datasets. Moreover, considering ET dependency on LST accuracy, the DMS sharpener together with Sentinel-2 and -3 cloud masks shall be enriched. Even though a more rigorous cloud layer for the TIR sensor is under active debate among scientists (*https://thermal2023.esa.int/*), the most recent S3 SLSTR Validation Report comes from 2017 providing unimproved data. Considering irregular ET observations affected by cloudy-sky conditions, future research should also concentrate on a data-driven approach to recover overcast cell grids. Furthermore, to capture time-induced characteristics of the land surface, LST downscaling, and TSEB-PT models might be additionally driven by more frequent variables rather than 10-day Sentinel-2 reflectances. The temporal mismatch between S2 and S3 can be solved by applying Harmonized Landsat-Sentinel imagery, which in parallel minimizes the probability of cloudiness. Consequently, due to some inaccuracies in the Sen2Cor cloud mask, other products like Sen2cloudless and Fmask shall be integrated into the workflow. The enhanced products would be a valuable tool to accurately estimate the components of the terrestrial water cycle on a regional scale and over heterogeneous ecosystems.

# Appendices

## Appendix A

**Fig. A1: Locations of the Eddy Covariance towers used for validating the ET product at (a) IT-MtM (source:** *https://browser.lter.eurac.edu/*)**, (b) IT-MtP (source:** *https://browser.lter.eurac.edu/*)**, (c) IT-Lsn (source:** *https://www.icos-italy.it/lison-it-lsn/*)**, (d) IT-Tor (source:** *https://www.icos-italy.it/elementor-2052/*)**, (e) IT-Ren (source:** *https://deims.org/5d32cbf8-ab7c-4acb-b29f-600fec830a1d*)**, (f) FR-Pue (source:** *https://meta.icos-cp.eu/resources/stations/ES_FR-Pue*)**, (g) IT-SR2 (source:** *https://meta.icos-cp.eu/resources/stations/ES_IT-SR2*)**, (h) IT-MBo (source:** *Sakowska et al., 2014*)** sites.**

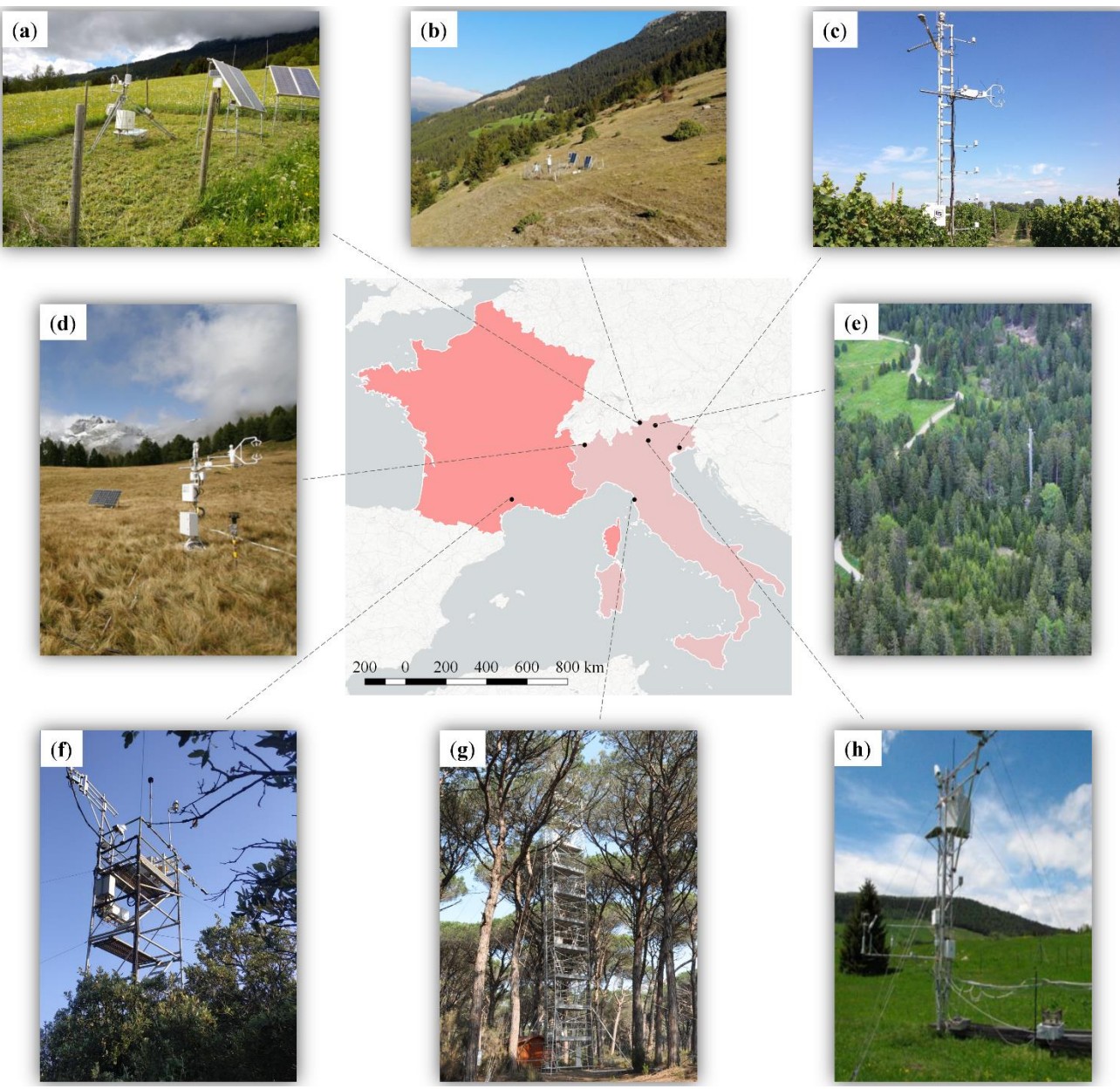

## Appendix B

**Tab. B1: Site-based accuracy results for ET product obtained from 1-km Sentinel-3 LST and its corresponding downscaled version at 100 m spatial resolution. Note that for some sites low resolution ET grids are not available due to cloud coverage detected in TSEB-PT input variables within 1-km pixels. For this reason, three sites (i.e., IT-SR2, IT-MtP) were not included in this analysis.**

| Flux site | RMSE | | R | | MB | |
|---|---|---|---|---|---|---|
| | 1 km | 100 m | 1 km | 100 m | 1 km | 100 m |
| IT-Tor | 2.00 | 1.89 | 0.44 | 0.38 | -1.52 | -1.51 |
| IT-Lsn | 0.84 | 0.73 | 0.87 | 0.94 | -0.25 | 0.48 |
| IT-MBo | 1.40 | 0.98 | 0.48 | 0.67 | -0.79 | -0.28 |
| IT-MtM | 2.80 | 2.59 | 0.49 | 0.39 | -2.06 | -1.47 |
| IT-Ren | 3.08 | 2.35 | 0.72 | 0.94 | -2.70 | -1.64 |
| FR-Pue | 1.90 | 2.13 | -0.16 | -0.15 | 0.85 | 0.99 |

## Appendix C

**Fig. C1: Spatial coverage of Sentinel-2 tiles together with their overlapping areas for (a) Po, (b) Ebro, (c) Medjerda, (d) Hérault basins. Each tile (in green) contains respective identification number corresponding to Sentinel-2 tiling system as explained at *https://sentinels.copernicus.eu/web/sentinel/missions/sentinel-2/data-products*.**

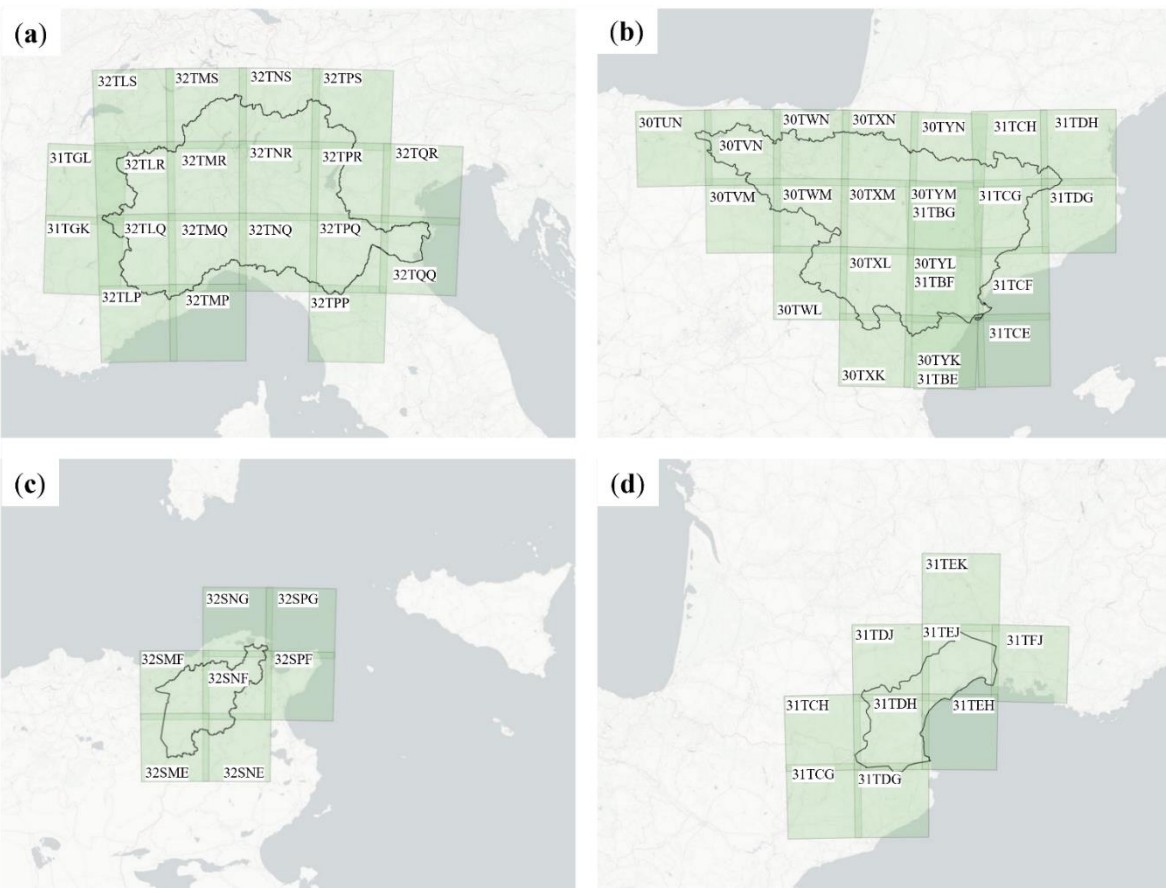

## Author contribution

MC and PB contributed to the research conceptualization and its implementation. Data collection, processing, and data analysis were conducted by PB with the assistance of BV. AJ and BV were responsible for the cloud computing infrastructure setup and the database management. MC supervised and supported the entire work. The original draft was prepared by PB, and then 715 revised in collaboration with MC, BV, and AJ.

## Competing interests

The authors of this paper declare no conflict of interest.

## Acknowledgements

This work was supported by European Space Agency in the framework of 4DMED-Hydrology project: "Developing an 720 advanced, high-resolution, and consistent reconstruction of the Mediterranean terrestrial water cycle" (ESA Contract No. 4000136272/21/I-EF). The authors would like to thank Petra Hulsman and Diego Miralles for preparing and sharing pre-processed *in-situ* data from Eddy Covariance towers.

## Abbreviations and acronyms

The following abbreviations and acronyms are used in this work:

| | |
|---|---|
| $\alpha_{PT}$ | Priestley-Taylor coefficient for potential transpiration (default = 1.26) |
| CC | Cloud cover |
| CERRA | Copernicus European Regional ReAnalysis |
| COG | Cloud Optimized GeoTIFF |
| DMS | Data Mining Sharpener |
| DT | Decision Trees |
| ECMWF | European Centre for Medium-Range Weather Forecasts |
| ECOSTRESS | ECOsystem Spaceborne Thermal Radiometer Experiment |
| EFDC | European Fluxes Database Cluster |
| EO | Earth Observation |
| EODC | Earth Observation Data Centre |
| ERA5 | European ReAnalysis V5 |
| ESA | European Space Agency |
| ESVEP | End-member-based Soil and Vegetation Energy Partitioning |
| ET | Evapotranspiration |
| FAO | Food and Agriculture Organization |
| GLDAS | Global Land Data Assimilation System |
| GLEAM | Global Land Evaporation Amsterdam Model |
| GPT | Graphical Processing Tool |
| HR | High resolution |
| ICOS | Integrated Carbon Observation System |
| IQR | Interquartile range |

| | |
|---|---|
| **LAI** | Leaf Area Index |
| **LC** | Landcover |
| **LSA-SAF** | Land Surface Analysis - Satellite Application Facility |
| **LST** | Land Surface Temperature |
| **LSTM** | Copernicus Land Surface Temperature Monitoring |
| **METRIC** | Mapping Evapotranspiration with Internalized Calibration |
| **MODIS** | Moderate Resolution Imaging Spectroradiometer |
| **MR** | Mediterranean Region |
| **MSG** | Meteosat Second Generation |
| **MSI** | MultiSpectral Instrument |
| **NASA** | National Aeronautics and Space Administration |
| **OLCI** | Ocean and Land Colour Instrument |
| **PDU** | Product Dissemination Unit |
| **PROBA-V** | Project for On-Board Autonomy - Vegetation |
| **Sen-ET** | Sentinels for Evapotranspiration |
| **SEVIRI** | Spinning Enhanced Visible and InfraRed Imager |
| **SGB** | Surface Biology and Geology |
| **SLSTR** | Sea and Land Surface Temperature Radiometer |
| **S-NPP** | Suomi National Polar-orbiting Partnership |
| **SR** | Surface Reflectance |
| **SRTM** | Shuttle Radar Topography Mission |
| **STAC** | Spatio Temporal Asset Catalogue |
| **TIR** | Thermal Infrared |
| **TRISHNA** | Thermal infraRed Imaging Satellite for High-resolution Natural resource Assessment |
| **(T)SEB** | (Two-)Source Energy Balance |
| **VIIRS** | Visible Infrared Imaging Radiometer Suite |
| **VSWIR** | Visible ShortWave InfraRed |
| **WaPOR** | Water Productivity through Open access of Remotely sensed derived data |
| **WGS84** | World Geodetic System 1984 |

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
