# Peer review of "A Copernicus-based evapotranspiration dataset at 100-m spatial resolution over four Mediterranean basins"

_Earth System Science Data, 2023_

## Author Comment (AC1)

**RC1**: 'Comment on essd-2023-466', William Kustas, 27 Mar 2024

The paper by Bartkowiak et al on the application of remote sensing-based energy balance model (TSEB-PT) in the Mediterranean region is well written with a robust scientific approach and analysis. However, there are two areas that the authors need to address in order for the reader to have a better understanding of the uncertainty in both the model and measurements. It appears the authors have chosen several challenging sites (some in complex topography) to conduct their model application and validation. This requires them to discuss in greater detail the measurement uncertainty. For example, they should make mention of the kind of energy balance closure they obtain at the different sites and if they used some method to force closure. Merely providing a reference to the processing of the eddy covariance data isn't sufficient for the reader to easily interpret these results. In addition, for sites with sloping/complex terrain do they know if a planar fit was incorporated in post-processing the eddy covariance measurements (e.g., Ross and Grant, 2015)? Do the sites with complex topography have worse energy balance closure than more flat terrain? If so, this could factor into larger scatter observed at those sites. Finally, TSEB was not originally developed to be applied in complex terrain, although ways to incorporate refinements to TSEB for complex terrain is a worthwhile endeavor and should be mentioned.

I would also like to draw their attention to other studies that have been able to find better results over forested sites, although still a tendency for larger scatter (Hadi et al., 2022). Others have accounted for the green fraction from remote sensing and PT alpha term from knowledge of land cover (Guzinski et al., 2013; Andreu et al., 2018). Of course one may consider adjusting the PT alpha term a kind of tuning, but I am sure the authors are aware that land cover information should be used wherever possible since knowledge of the land cover type factors into a number of the TSEB-PT model parameters. Finally, there are other studies applying the multiscale version of TSEB that have obtained good results over pine forests (Yang et al 2017; 2020). Although the authors do point out that some of the sites are challenging, I think they should also reference work that suggests applications of TSEB over forested areas can achieve reasonable results, especially when the surface is not complex.

References:

Andreu, A., Kustas, W.P., Polo, M.J., Carrara, A., & González-Dugo, M.P. (2018). Modeling surface energy fluxes over a dehesa (oak savanna) ecosystem using a thermal based two-source energy balance model (TSEB) I. *Remote Sensing, 10,* doi:10.3390/rs10040567

Guzinski, R., Anderson, M.C., Kustas, W.P., Nieto, H., & Sandholt, I. (2013). Using a thermal-based two source energy balance model with time-differencing to estimate surface energy fluxes with day-night MODIS observations. *Hydrol. Earth Syst. Sci., 17*, 2809-2825

Jaafar, H.H., Mourad, R.M., Kustas, W.P., & Anderson, M.C. (2022). A global implementation of single and dual-source surface energy balance models for estimating actual evapotranspiration at 30-m resolution using Google Earth Engine. *Water Resour. Res.*, 58, doi:10.1029/2022WR032800.

Ross, A.N. and Grant, E.R. (2015) A new continuous planar fit method for calculating fluxes in complex, forested terrain *Atmos. Sci. Let., 16,* 445–452

Yang, Y., Anderson, M.C., Gao, F., Hain, C.R., Semmens, K.A., Kustas, W.P., Normeets, A., Wynne, R.H., Thomas, V.A., & Sun, G. (2017). Daily Landsat-scale evapotranspiration estimation over a managed pine plantation in North Carolina, USA using multi-satellite data fusion. *Hydrol. Earth Syst. Sci., 21*, 1017-1037

Yang, Y., Anderson, M., Gao, F., Hain, C., Noormets, A., Sun, G., Wynne, R., Thomas, V., & Sun, L. (2020). Investigating impacts of drought and disturbance on evapotranspiration over a forested landscape in North Carolina, USA using high spatiotemporal resolution remotely sensed data. *Remote Sens. Environ., 238*, 111018

Dear William, thank you very much for your review and useful comments.

First, we would like to explain the measurements uncertainty at the validation sites. Indeed, when all input parameters were available, we have computed the energy balance closure (EBC) ratio (i.e., [Rn-G-H]/LE with Rn: net radiation, G: soil heat flux, H: sensible heat flux, LE: latent heat flux) for the EC sites. In total, the EBC values are derived for five flux sites in Italy. The obtained ratios are as follows:

- vineyard at IT-Lsn (1 m a.s.l.): 0.28,
- grassland at IT-MtM (1450 m a.s.l.): 0.08,
- grassland at IT-MtP (1550 m a.s.l.): 0.13,
- evergreen broadleaf forest at IT-SR2 (4 m a.s.l.): 0.73,
- grassland at IT-Tor (2160 m a.s.l.): -0.02.

After contacting providers of daily flux data from University of Ghent in Belgium, we were recommended to include all eight sites in our analysis due to the small number of EC towers over our study areas. We derived the largest ratio over lowland forest at IT-SR2 exceeding 0.7, while EBC values over sloping terrain (i.e., IT-MtM, IT-MTP, and IT-Tor) achieved acceptable scores (i.e., below 0.2). Unfortunately, for the remaining three sites the EBC ratios were not provided (i.e., FR-Pue, IT-Ren, IT-MBo). As can be seen above, in our case, the energy balance closure values do not depend on topographic complexity. On the other side, to make any conclusions in this regard, we truly believe that the impact of complex topography on energy balance closure shall be further investigated by incorporating more flux sites across different landcovers, topographies, and (micro)climates. We contacted PIs of the EC towers to understand if they applied a planar fit method to eddy covariance flux data. We have received their feedback regarding sites over forested landscapes at FR-Pue, IT-Ren and IT-SR2, and high-mountain grasslands located in Aosta Valley (IT-Tor) and Mazia Valley (IT-MtM and IT-MtP). The first two tree covered sites (FR-Pue, IT-Ren) were processed using a double rotation method, while a planar fit method was applied to IT-SR2. Regarding the alpine grasslands, all eddy covariance flux sites were corrected with a planar fit approach. In addition to the research work of Ross and Grant (2015), we have included ancillary studies investigating EC measurement uncertainties:

1. Castelli, M., Anderson, M.C., Yang, Y., Wohlfahrt, G., Bertoldi, G., Niedrist, G., Hammerle, A., Zhao, P., Zebisch, M. and Notarnicola, C., 2018. Two-source energy balance modeling of evapotranspiration in Alpine grasslands. Remote Sensing of Environment, 209, pp.327-342.

2. Mauder, M., Cuntz, M., Drüe, C., Graf, A., Rebmann, C., Schmid, H.P., Schmidt, M. and Steinbrecher, R., 2013. A strategy for quality and uncertainty assessment of long-term eddy-covariance measurements. Agricultural and Forest Meteorology, 169, pp.122-135.

3. Rannik, Ü., Vesala, T., Peltola, O., Novick, K.A., Aurela, M., Järvi, L., Montagnani, L., Mölder, M., Peichl, M., Pilegaard, K. and Mammarella, I., 2020. Impact of coordinate rotation on eddy covariance fluxes at complex sites. Agricultural and Forest Meteorology, 287, p.107940.

Description on measurements uncertainties was added in the revised version of the manuscript in the section 2.2 (lines 189-193), and also it has been discussed in the 'Results and discussion' (lines 479-482) and in the Conclusions (lines 642-644).

Second, we fully agree with you that more attempts are required to enhance the TSEB performance over heterogenous terrain. This is a part of our current work where we aim to improve the model in the European Alps, including forest sites and areas with complex topography. In this regard, topographically corrected

solar radiation and landcover at finer spatial resolutions shall be incorporated to account for heterogenous landscapes and different types of vegetation along with their biophysical characteristics. Moreover, as mentioned by you before, the Priestely-Taylor parameter with a default value of 1.26 needs to be adjusted to take into account variations in green vegetation cover during growing season. All described above improvements are our close-future objectives in order to derive more reasonable results, especially over forests as shown in the suggested reference research studies. To improve our manuscript, in the section 4.1 (lines 485-497) we have added some text where together with the suggested reference papers we describe potential improvements in the TSEB-PT for deriving more robust ET estimates over forests and areas with complex topography. Additionally, in the revised manuscript we included the following research publication on TSEB modelling over boreal forests:

1. Cristóbal, J., Prakash, A., Anderson, M.C., Kustas, W.P., Alfieri, J.G. and Gens, R., 2020. Surface energy flux estimation in two Boreal settings in Alaska using a thermal-based remote sensing model. *Remote Sensing*, *12*(24), p.4108.

---

## Author Comment (AC2)

**RC2**: 'Comment on essd-2023-466', Anonymous Referee #2, 10 Apr 2024

Review of "A Copernicus-based evapotranspiration dataset at 100-m spatial resolution over the Mediterranean region" by Bartkowiak et al.

This paper presents an original evapotranspiration (ET) data set at 100 m resolution over 4 basins of the Mediterranean region: Ebro basin in Spain, Po basin in Italy, Herault basin in France and Medjerda basin in Tunisia. The main originality of the data set is its high spatial resolution compared to that of existing ET products classically available at 1 km or coarser resolution. The new 100 m resolution ET data set is derived by automatizing existing codes based on TSEB (Two-Source Energy Balance) model and Sentinel-2 (S2) and Sentinel-3 (S3) remote sensing data. The satellite-derived ET estimates are evaluated with eddy covariance measurements collected at 8 sites with 7 located in Italy and 1 in France and with several land covers (grassland, evergreen broadleaf forest, evergreen needleleaf forest and vineyard). Although the presented data set may be of interest for many different applications over the basins studied, I think that the evaluation strategy must be improved to really demonstrate the better accuracy of the new data set.

We would like to thank the reviewer for revising the manuscript and providing helpful suggestions and comments to improve this work. Below we provide answers to the reviewer's comments.

I recommend major revisions taking into account the concerns listed below:

1) Title, abstract and conclusion: The extent of the data set is confusing and somehow over-sold as the actual data set does not cover the entire Mediterranean region but only four selected basins within the Mediterranean region. It is true that the algorithms developed by the authors should work over other parts of the Mediterranean region but the paper focuses on the dataset. The authors should be more specific in the title, abstract and conclusion.

**Response 1:** Thank you very much for this useful comment. Indeed, it is true that our data paper focuses on 100-m evapotranspiration estimation over selected parts of the Mediterranean region. In order to avoid confusion regarding dataset extent, we have modified the title and the Conclusions section. In the abstract we directly indicate our study areas (i.e., Ebro, Po, Herault, and Medjerda basins) where ET is produced.

2) Evaluation of the 100 m resolution evapotranspiration dataset: The evaluation of satellite-derived evapotranspiration estimates is generally sound. However in my opinion it suffers from two major weaknesses. As outlined in the abstract and introduction and other parts of the paper, the rationale for developing a new ET product at high spatial resolution is that common products available at coarser spatial resolution are not sufficient to characterize the very high heterogeneity of land surfaces. The validation strategy of their product should support this key point. This is all the more needed as the ET product relies on the downscaling of 1 km resolution S3 land surface temperature (LST) data from 1 km to 100 m resolution. The evaluation of 100 m satellite-derived ET must be consolidated by estimating the gain in accuracy provided by the use of 100 m resolution remote sensing data, instead of 1 km resolution remote sensing data. One way of achieving this would be to implement PT-TSEB at 1 km resolution at the validation sites and calculate performance metrics as is done for the 100 m resolution dataset. Another drawback of the validation exercise is that it is based on only 8 stations, with 7 located in the same (Po) basin. Readers need to be convinced that this data set is significantly better than other more classical data sets.

The rationale for developing a new ET product at high spatial resolution:

Line 9-10: "existing global products with spatial resolution >=0.5 km are insufficient to capture spatial detail at a local level"

Line 239: "the Mediterranean region characterized by complex topography and highly patched landcover, where 1-km ET maps might not fully represent spatial heterogeneities of the land surface"

**Response 2:** We fully agree with your point that the TSEB performance driven by both original Sentinel-3 LST data and its 100-m downscaled product shall be carried out in order to evaluate the effectiveness of thermal sharpening on ET retrievals. In this work, we have exploited data mining sharpener (DMS) of Gao et al. (2012) successfully used in many research studies for estimating high spatial resolution TIR-ET (Anderson et al., 2021; Guzinski and Nieto, 2019; Yang et al., 2021; Guzinski et al., 2023). As presented by *Guzinski and Nieto* (2019), TSEB-PT driven by downscaled DMS-based surface temperatures is more performant compared to ET estimates driven by original 1-km LST data with around a 13% increase in Pearson correlation coefficient (r) between in-situ ETs and their corresponding modelled observations. Furthermore, the authors of the ESA Sen-ET report estimated evapotranspiration using METRIC, ESVEP, and TSEB-PT algorithms at 11 flux tower sites across different vegetation types and climate zones, and derived the best accuracy scores from the latter model when data mining LST sharpener (either based on Artificial Neural Networks or Decision Trees regressors) was applied ([https://www.esa-sen4et.org/downloads/prototype_evaluation_v1.3.pdf](https://www.esa-sen4et.org/downloads/prototype_evaluation_v1.3.pdf)). According to the report, the Priestley-Taylor Two-Source Energy Balance of ET was ranked as the most robust approach with consistently lower Root Mean Square Error (RMSE) and higher correlation for latent flux yielding an average RMSE of 90 W m$^{-2}$ and r exceeding 0.7, which largely outperforms METRIC and ESVEP by more than 11% and 30% for RMSE and Pearson correlation, respectively. Furthermore, the TSEB-PT has been constantly updated in order to improve the modelling scheme for thermal sharpening, and as reported in Guzinski et al. (2023) enhanced DMS-driven TSEB-PT at field scale achieved accuracy of 0.8 mm per day, which is our next-future goal to be implemented. Moreover, Sánchez et al. (2023) conducted extensive study on the performance of LST downscaling in Spain, and based on their validation results with in-situ measurements the DMS approach gave nearly two times smaller RMSE error compared to the 1-km S3 LST. In addition to the abovementioned literature review, in our co-authored paper we compared Sen-ET outcomes with other evapotranspiration products, including 3-km MSG SEVIRI and 70-m ECOSTRESS ET which on average gave less robust accuracy metrics than our 100-m retrievals (De Santis et al., 2022). These results and other authors' findings moved us towards generation of 100-m ET dataset. Therefore, we have decided to apply the LST sharpening strategy in our ET workflow assuming its better performance in different land covers and climates compared to original 1-km S3-driven TSEB-PT. In the section 3.2 of the revised manuscript together with relevant research papers we provide more information on the performance of DMS procedure for estimating high spatial resolution evapotranspiration (lines 327-340; 358-363).

Considering the scarcity of eddy covariance towers over the Mediterranean catchments and time of interest (2017-2021) for our analysis, together with University of Ghent we have managed to gather in-situ observations at only eight eddy covariance sites (i.e., seven locations in Italy and one site in France) that provide long time-series of latent heat flux. In order to get more general conclusions on the results, we fully agree that the validation shall be performed including more in-situ EC towers represented by wider variety of landcover types, climate zones, and topography which is our future priority objective. This might be done by extending the spatial coverage of the ET data in order to increase number of available local flux measurements.

1. Gao, F., Kustas, W. P. and Anderson, M. C.: A data mining approach for sharpening thermal satellite imagery over land. Remote Sensing, 4(11), pp.3287-3319, 2012.
2. Anderson, M. C., Yang, Y., Xue, J., Knipper, K. R., Yang, Y., Gao, F., Hain, C. R., Kustas, W. P., Cawse-Nicholson, K., Hulley, G. and Fisher, J. B.: Interoperability of ECOSTRESS and Landsat for mapping evapotranspiration time series at sub-field scales. Remote Sensing of Environment, 252, p. 112189, 2021.

3.  Guzinski, R. and Nieto, H.: Evaluating the feasibility of using Sentinel-2 and Sentinel-3 satellites for high-resolution evapotranspiration estimations. Remote sensing of Environment, 221, pp.157-172, 2019.
4.  Yang, Y., Anderson, M. C., Gao, F., Wood, J. D., Gu, L. and Hain, C.: Studying drought-induced forest mortality using high spatiotemporal resolution evapotranspiration data from thermal satellite imaging. Remote Sensing of Environment, 265, p.112640, 2021.
5.  Guzinski, R., Nieto, H., Sánchez, R. R., Sánchez, J. M., Jomaa, I., Zitouna-Chebbi, R., Roupsard, O. and López-Urrea, R.: Improving field-scale crop actual evapotranspiration monitoring with Sentinel-3, Sentinel-2, and Landsat data fusion. International Journal of Applied Earth Observation and Geoinformation, 125, p.103587, 2023.
6.  Sánchez, J. M., Galve, J. M., Nieto, H. and Guzinski, R.: Assessment of High-Resolution LST Derived From the Synergy of Sentinel-2 and Sentinel-3 in Agricultural Areas. IEEE Journal of Selected Topics in Applied Earth Observations and Remote Sensing, 17, pp. 916-928, 2023.
7.  De Santis, D., D'Amato, C., Bartkowiak, P., Azimi, S., Castelli, M., Rigon, R. and Massari, C.: Evaluation of remotely-sensed evapotranspiration datasets at different spatial and temporal scales at forest and grassland sites in Italy. In 2022 IEEE Workshop on Metrology for Agriculture and Forestry (MetroAgriFor) (pp. 356-361). IEEE, November 2022.

3) Introduction:

- Second paragraph of the introduction: when the authors review existing evapotranspiration models, they mention process-based (energy balance models) and data-driven (statistical models) approaches. The so-called contextual/semi-empirical approaches are missed. I recommend completing this state of the art by adding a few references to contextual methods.

**Response 3a:** Thank you for your comment. We have included the contextual methods for evapotranspiration retrieval. You can find our modifications in the second paragraph of the Introduction section (lines 46-49). In the revised version of the manuscript, we included some research papers on the contextual ET methods. They are as follows:

1.  Bastiaanssen, W. G. M., Noordman, E. J. M., Pelgrum, H., Davids, G., Thoreson, B. P. and Allen, R. G.: SEBAL model with remotely sensed data to improve water-resources management under actual field conditions. Journal of irrigation and drainage engineering, 131(1), pp. 85-93, 2005.
2.  Chirouze, J., Boulet, G., Jarlan, L., Fieuzal, R., Rodriguez, J. C., Ezzahar, J., Er-Raki, S., Bigeard, G., Merlin, O., Garatuza-Payan, J. and Watts, C.: Intercomparison of four remote-sensing-based energy balance methods to retrieve surface evapotranspiration and water stress of irrigated fields in semi-arid climate. Hydrology and earth system sciences, 18(3), pp. 1165-1188, 2014.
3.  Sobrino, J. A., Souza da Rocha, N., Skoković, D., Suélen Käfer, P., López-Urrea, R., Jiménez-Muñoz, J. C. and Alves Rolim, S. B.: Evapotranspiration Estimation with the S-SEBI Method from Landsat 8 Data against Lysimeter Measurements at the Barrax Site, Spain. Remote Sensing, 13(18), p. 3686, 2021.
4.  Trezza, R., Allen, R. G. and Tasumi, M.: Estimation of actual evapotranspiration along the Middle Rio Grande of New Mexico using MODIS and landsat imagery with the METRIC model. Remote Sensing, 5(10), pp. 5397-5423, 2013.

- Line 96: "many data-driven approaches have been proposed, relying on empirical relationships between 1-km surface temperatures and high-resolution explanatory variables derived from Synthetic Aperture Radar (SAR) and Visible Shortwave Infrared (VSWIR) sensors (Li et al. 2019; Mao et al., 2021; Pu and Bonafoni, 2023)." As none of the above references include SAR data I suggest this one: Amazirh et al.

2019. Including Sentinel-1 radar data to improve the disaggregation of MODIS land surface temperature data. ISPRS journal of photogrammetry and remote sensing, 150, 11-26.

**Response 3b:** Thank you very much for this suggestion. We have included this interesting research paper as a reference in the manuscript (line 102).

4) Eddy covariance measurements:

I could find no information on how the authors derived daily ET estimates from 30-min eddy covariance measurements. Equation (1) explains the unit conversion from W/m2 to mm/day, but the aggregation of hourly eddy covariance measurements at the daily scale is not described at all (?).

**Response 4:** To be more precise with the description for daily ET aggregation procedure from 30-min ground observations at available EC towers, in the third paragraph of the section 2.2 (starting from line 184) we provide more detailed information on daily ET retrieval according to the strategy developed by the project partners from the University of Ghent working in hydrology domain. Additionally, we provide a reference paper of Hulsman et al. (2023) that explains the preprocessing procedure for the in-situ EC observations:

1. Hulsman, P., Keune, J., Koppa, A., Schellekens, J. and Miralles, D. G.: Incorporating Plant Access to Groundwater in Existing Global, Satellite-Based Evaporation Estimates. Water Resources Research, 59(8), p. e2022WR033731, 2023.

5) Spatio-temporal coverage of the dataset:

I am surprised by the relatively large and frequent gaps in the ET dataset due to cloud cover. I imagine that the S2 dataset composited over 10 days and the S3 dataset composited over 10 days separately have greater spatial coverage. I wonder if the relatively low spatio-temporal coverage of the ET dataset is associated with the temporal mismatch between S2 and S3 overpasses?

**Response 5:** As described in the Methodology section (lines 248-253; 306-309), we have generated daily ET maps with spatio-temporal coverage corresponding to daily Sentinel-3 (S3) LST data and 10-day Sentinel-2 composite product specially adjusted to S3 acquisition days. Indeed, in case of Sentinel-2 we minimized the cloud occurrence by means of temporal compositing, while S3 LST is more affected by overcast conditions. Sentinel-3 LST datasets were not composited and this is the main reason for relatively large spatiotemporal gaps in the daily ET product.

6) TSEB modeling

One of the difficulties in spatializing the TSEB over large areas is characterizing the aerodynamic resistance (linked to canopy height, leaf size, etc.) and the green component fg of the vegetation cover. Can you briefly present the range of values chosen for these key parameters for the main vegetation types in the basins studied?

**Response 6:** Thank you for this comment. Canopy aerodynamic resistance (Rx) and green vegetation cover (fg) are expressed as follows:

1. $Rx = (C'/F) * [l_w/(u_{do}+z_{om})]^{1/2}$

where C´ is derived from weighting a coefficient in the formulation for leaf boundary layer resistance over the height of canopy (it assumed to be equal to 90 $s^{1/2}\,m^{-1}$), F is local leaf area index (i.e., LAI/fc with fc: fractional vegetation cover), lw is the effective leaf width, and $u_{do}+z_{om}$ corresponds to the wind speed within the canopy-air interface (Norman et al., 1995).

2.  $fg = FAPAR/FIPAR$

where FAPAR is the fraction of absorbed photosynthetically active radiation obtained from the ESA Snap biophysical processor, and FIPAR corresponds to the fraction of photosynthetically active radiation intercepted by green and brown vegetation and it is expressed using following formula:

3.  $FIPAR = 1 - exp[-0.5*PAI/cos\Theta]$

where PAI = LAI / fg.

As shown above, the retrieval of fg is an iterative procedure that re-calculates FIPAR parameter until fg converges (Guzinski et al., 2020).

Green vegetation cover is driven by plant area index (PAI), FAPAR, and sun zenith angle ($\Theta$) derived from Sentinel-2 reflectance imagery. It means that fg values are estimated for each S2 pixel in space and time ranging from 0 to 1. Similarly to fg product, aerodynamic resistance at the canopy boundary layer is based on Sentinel-2 grid and changes in time since it is derived from Earth Observation inputs, such as LAI, ERA5 wind speed, and landcover information derived from ESA CCI LUT (e.g. lw) following Guzinski et al. (2019).

1.  Norman, J. M., Kustas, W. P. and Humes, K. S., 1995. Source approach for estimating soil and vegetation energy fluxes in observations of directional radiometric surface temperature. Agricultural and Forest Meteorology, 77(3-4), pp.263-293.
2.  Guzinski, R. and Nieto, H., 2019. Evaluating the feasibility of using Sentinel-2 and Sentinel-3 satellites for high-resolution evapotranspiration estimations. Remote sensing of Environment, 221, pp.157-172.
3.  Guzinski, R., Nieto, H., Sandholt, I. and Karamitilios, G., 2020. Modeling high-resolution current evapotranspiration through Sentinel-2 and Sentinel-3 data fusion. Remote Sensing, 12 (9), p.1433.

7) Correction of input meteorological data for topography effects:

Line 337: "All extracted variables from the reanalysis dataset, except for wind speed, are corrected for terrain using the SRTM DEM product"
Line 440: "The distribution of solar radiation, wind speed, and air temperature gradients are less influenced by a landscape complexity over mountain plateau than over steep slopes, and thus coarse resolution ERA5 might be more representative for … "
Line 474: "The ET models are controlled by climate inputs derived from 31-km fields…"

The above statements seem contradictory. Can you please describe how solar radiation and air temperature are downscaled at 100 m resolution using the DEM? Both variables have a very strong effect especially in areas of complex topography such as the basins studied?

**Response 7:** Indeed, air temperature and solar radiation from ERA5 data were enhanced with SRTM DEM. While shortwave radiation was corrected for illumination conditions using elevation, air temperature (TA) originally derived at 2-m height was recalculated to the blending height of 100 m using the elevation product and standard lapse rate of 6.5 K/1000 m. The blending height for low-spatial resolution air temperature is assumed to be more representative rather than TA at the height of 2 m due to weaker impact of land-atmosphere interactions at 100-m above ground surface. We expect a strong impact of those variables over

complex areas, such as mountains where most EC towers are located in this study. To be more consistent, we have added some text (lines 234-235; 365-370) to explain the utility of DEM in the ERA5 input data adjustments together with a relevant research paper:

1. Guzinski, R., Nieto, H., Sánchez, J. M., López-Urrea, R., Boujnah, D. M. and Boulet, G.: Utility of copernicus-based inputs for actual evapotranspiration modeling in support of sustainable water use in agriculture. IEEE Journal of Selected Topics in Applied Earth Observations and Remote Sensing, 14, pp.11466-11484, 2021.

8) Shadows effects

Line 450: «The poor accuracy at forested sites might be related to their complex tree structures and multilayer composition which is not considered in Sen-ET». Since the authors are evaluating their product at Puechabon site, I suggest referring to Penot et al. (2023). Estimating the water deficit index of a Mediterranean holm oak forest from Landsat optical/thermal data: a phenomenological correction for trees casting shadow effects. IEEE Journal of Selected Topics in Applied Earth Observations and Remote Sensing.

**Response 8:** Thank you very much for this suggestion. We included this publication in our manuscript (line 485).

9) Discussion of sources of uncertainty in the ET dataset (Lines 465-473)

Another source of uncertainty that should be mentioned and discussed is the intrinsic limitation of downscaling methods of LST data using reflectances as high resolution ancillary information.

**Response 9:** We fully agree with the reviewer's comment. We have included this aspect in the indicated paragraph of the revised manuscript (lines 510-514). We also added some reference papers to explain limitations of reflectance bands as predictors for thermal downscaling as depicted below:

1. Hu, Y., Tang, R., Jiang, X., Li, Z.L., Jiang, Y., Liu, M., Gao, C. and Zhou, X., 2023. A physical method for downscaling land surface temperatures using surface energy balance theory. Remote Sensing of Environment, 286, p. 113421.
2. Merlin, O., Duchemin, B., Hagolle, O., Jacob, F., Coudert, B., Chehbouni, G., Dedieu, G., Garatuza, J. and Kerr, Y.: Disaggregation of MODIS surface temperature over an agricultural area using a time series of Formosat-2 images. Remote Sensing of Environment, 114(11), pp. 2500-2512, 2010.

Edits:

- Once defined, acronyms must be used systematically (this problem appears in many places in the text). Also acronyms should be defined only once.
- There is an acronym for Languedoc Roussillon but not for the study basin (Herault basin)?
- Composted/composting (line 17, fin 2, 396, line 486)
- Unit is missing for RMSEs at line 584

**Response to edits:** Thank you. We have corrected the manuscript considering the abovementioned issues. In addition, we updated the table with a list of abbreviations and acronyms (page 27-28).

---

## Author Response (AR2)

The authors have put considerable efforts into addressing all my comments.

I am fully satisfied with most of their answers. However, I still have a problem with the validation exercise, which I still find incomplete (comment #2). The rationale for developing a new ET product at high spatial resolution is that common products available at coarser spatial resolution are not sufficient to characterize the very high heterogeneity of land surfaces. The validation strategy should therefore support this key point. In their response, the authors rely on a number of articles to support the better accuracy of their modeling approach at high resolution, but they do not provide any quantitative assessment using the in situ measurements available in this study. To better convince readers of the usefulness of this product, I would expect a comparison between 1 km resolution (S3-derived) ET estimates and in situ measurements. This would enable us to assess the performance of the high-resolution product in relation to more conventional products. Otherwise, is a validation exercise necessary after all?

**Response 1:**

Thank you very much for your comment. In order to investigate the impact of 1-km Sentinel-3 on final ET estimates, we rerun TSEB-PT using original LST at the flux sites used in this study, and then we compare validation results obtained from 100-m and 1-km ET simulations. On average, the resulting outcomes demonstrate an accuracy improvement when the model is forced by downscaled land surface temperature at 100 m spatial resolution. In fact, RMSE error dropped by around 13% and we observed a 12 % increase in R when high resolution LST was incorporated in TSEB-PT. We included this analysis and added some new text and tables in the revised version of the manuscript (lines 365-368; 432-434; 505-518; 648-651; 702-705).

Regarding comment #7, since the authors' response refers to Guzinski et al. (2021), I reread that paper and realized that the correction for topographic effects of solar radiation and air temperature was actually applied using a DEM at 300 m resolution. Is the same approach used in this study? If so, I think it is worth reminding this point in a revised version, as the target resolution (100 m) of the data set is considerably finer than that of the DEM used for topographic corrections of input meteorological data.

**Response 2:**

Thank you for pointing this out. In the revised version of the manuscript we refer to Guzinski et al. (2021) to give some information on the method for correcting topographic effects. Nevertheless, in our study we use high-resolution DEM obtained from Shuttle Radar Topography Mission at 90 m spatial resolution (see Table 2 in the revised manuscript on page 8).